# ALTER: All-in-One Layer Pruning and Temporal Expert Routing for Efficient Diffusion Generation

**Xiaomeng Yang**[1*], **Lei Lu**[1*], **Qihui Fan**[1], **Changdi Yang**[1], **Juyi Lin**[1],
**Yanzhi Wang**[1], **Xuan Zhang**[1], **Shangqian Gao**[2†]
[1]Northeastern University   [2]Florida State University
[1]{yang.xiaome, lu.lei1, fan.qih, yang.changd, lin.juy, yanz.wang,
xuan.zhang}@northeastern.edu   [2]sgao@cs.fsu.edu

## Abstract

Diffusion models have demonstrated exceptional capabilities in generating high-fidelity images. However, their iterative denoising process results in significant computational overhead during inference, limiting their practical deployment in resource-constrained environments. Existing acceleration methods often adopt uniform strategies that fail to capture the temporal variations during diffusion generation, while the commonly adopted sequential *pruning-then-fine-tuning strategy* suffers from sub-optimality due to the misalignment between pruning decisions made on pretrained weights and the model's final parameters. To address these limitations, we introduce **ALTER**: **A**ll-in-One **L**ayer Pruning and **T**emporal **E**xpoert **R**outing, a unified framework that transforms diffusion models into a mixture of efficient temporal experts. ALTER achieves a single-stage optimization that unifies layer pruning, expert routing, and model fine-tuning by employing a trainable hypernetwork, which dynamically generates layer pruning decisions and manages timestep routing to specialized, pruned expert sub-networks throughout the ongoing fine-tuning of the UNet. This unified co-optimization strategy enables significant efficiency gains while preserving high generative quality. Specifically, ALTER achieves same-level visual fidelity to the original 50-step Stable Diffusion v2.1 model while utilizing only 25.9% of its total MACs with just 20 inference steps and delivering a $3.64\times$ speedup through 35% sparsity.

## 1   Introduction

Diffusion models [1, 2, 3] emerged as a leading class of generative models, achieving quality results in diverse tasks such as image generation [4, 5, 6, 7], image editing [8, 9] and personalized content generation [10, 11]. However, they face application limits due to costly inference: repeated full-network denoising causes latency and memory use, restricting real-time and low-resource deployment.

To mitigate the high computational cost of diffusion models, two primary strategies have emerged: one focuses on reducing the number of denoising steps in the temporal dimension, and the other explores model-level compression to reduce the per-step computational burden. Extensive research has been devoted to the former, including improved samplers [12, 13, 14], distillation methods [15] and feature caching [16]. In this work, we focus on the latter: *how model compression can be further improved to enhance efficiency without sacrificing performance?* Pruning has been a long-standing approach for reducing model complexity, with recent advances focusing on structured pruning to enable hardware-friendly acceleration. Among various structured pruning schemes, layer-wise

---

* Equal Contribution
† Corresponding Author

39th Conference on Neural Information Processing Systems (NeurIPS 2025).

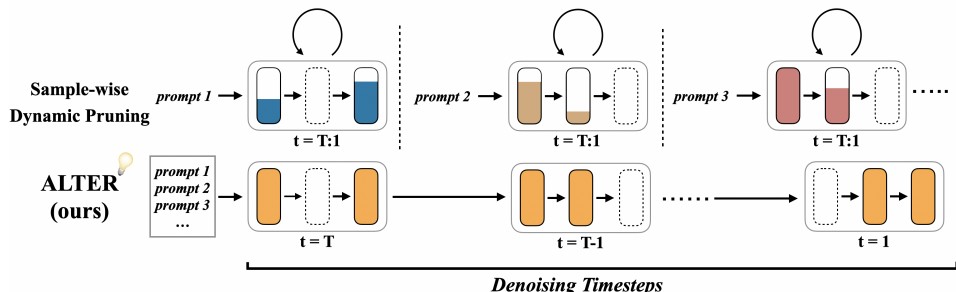

Figure 1: Comparison of model utilization in dynamic pruning. Sample-wise pruning can only use a static part of the model for one-specific image generation, while ALTER aims to achieve the full utilization of model capacity according to the necessity of each timestep.

pruning is the most practical in deployment, as it achieves speedup by discarding full layers with minimal modification to existing model execution pipelines. However, such coarse-grained pruning, especially when applied statically across all input prompts and denoising timesteps, often results in significant performance degradation due to both the complete removal of functional operators and the lack of flexibility to adapt to the varying computational demands.

To improve flexibility while retaining the deployment efficiency of layer-wise pruning, it is natural to combine dynamic pruning with coarse-grained structural removal, enabling different smaller models to be activated conditionally during inference. Existing efforts explore a sample-wise dynamic strategy, where a unique subnetwork is selected for each input [17, 18]. While this improves input adaptivity, it severely limits parameter utilization: *for a given sample, only a small subset of the model is used throughout the entire denoising trajectory.* To further unlock the potential of layer-wise pruning , we leverage the temporal asymmetry of diffusion, where different timesteps contribute unevenly to generation, to adaptively activate pruned substructures conditioned on the timestep, as shown in Fig. 1. This enables full model utilization across the trajectory, while suppressing redundant computation at each specific time step. While timestep-conditioned pruning improves overall parameter utilization, each substructure still requires fine-tuning to recover performance. Prior works often separate pruning and fine-tuning into distinct stages [19, 20], leading to a mismatch between fixed pruning decisions and evolving model weights. To address this, we propose to jointly optimize the pruning configuration and model parameters, allowing the substructures to co-adapt with the backbone during training.

To this end, we introduce the **A**ll-in-One **L**ayer Pruning and **T**emporal **E**xpert **R**outing (ALTER) framework, a unified and one-stage optimization approach that seamlessly integrates layer pruning with temporal expert routing and model fine-tuning. It effectively transforms the standard diffusion model into a mixture of efficient temporal experts, where each expert is a specialized pruned sub-network of the original model, tailored for distinct phases of the generation process. This dynamic configuration is achieved by identifying the optimal substructure for these different expert sub-networks and intelligently routing denoising timesteps to them throughout the entire finetuning phase. More specifically, we employ a hypernetwork to generate layer pruning decisions based on the updated model weights continuously, while managing the routing of timesteps to the appropriate experts at the same time. The impact of these pruning and routing decisions is simulated during training via a layer skipping mechanism in the forward pass. At inference time, the finalized hypernetwork allows for the selection of the most suitable expert and the skipping of designated layers for each timestep, thereby minimizing the temporal computational redundancy.

Our contributions can be summarized as follows:

**(1) We propose ALTER**: A novel framework that transforms diffusion models into a mixture of efficient temporal experts by unifying layer pruning, expert routing, and fine-tuning into a single-stage optimization process.

**(2) Temporal Expert Routing:** We introduce a hypernetwork-based mechanism that dynamically generates layer pruning decisions and manages timestep routing to specialized experts. Meanwhile, the shared denoising backbone is fully utilized by temporal experts to maximize parameter utilization.

**(3) Strong Empirical Results:** We conduct extensive experiments to show that ALTER significantly reduces computational costs (e.g., 25.9% FLOPs of 50-step Stable Diffusion v2.1) and accelerates inference (e.g., $3.64\times$ speedup with 20 steps) thanks to the nature of layer-wise pruning while maintaining the same-level generative fidelity.

## 2 Related Work

### 2.1 Efficient Diffusion Models

Efforts to enhance the efficiency of diffusion models primarily explore two avenues: (1) sampling acceleration, such as fast samplers [21, 22, 13, 14, 12, 23, 24], step distillation [15, 25, 26, 27, 28, 29] and feature caching [16, 30, 31, 32, 33]. (2) model compression via techniques like pruning, quantization [34, 35], or efficient attention mechanisms [36]. And our work focus on pruning to compress the structure. Pruning aims to reduce model size and inference latency with minimal performance loss [37, 38, 39, 40, 41, 42, 43]. Unstructured pruning [44, 45, 46, 47] removes individual weights, leading to sparse patterns challenging to accelerate on common hardware. In contrast, structured pruning [48, 49, 50, 51, 52] removes entire components (e.g., channels, layers), yielding more readily deployable speedups. Within diffusion models, early structured pruning efforts often applied static pruning decisions [53]. For instance, LAPTOP-Diff [19] and LD-Pruner [20] identify and discard less important UNet layers, offering coarse-grained but highly efficient static pruning. However, the optimal model structure can vary significantly across the iterative denoising process. More recent advances have explored dynamic pruning [54], where the model learns to prune in a data- or context-dependent manner for flexibility in resource-constrained inference scenarios. APTP [18], for example, incorporates a prompt-routing model that learns and adaptively routes inputs to appropriate sub-architectures. However, although APTP's sample-wise approach improves adaptivity over static pruning, it typically commits to a single sub-model for the entire denoising trajectory, which can lead to under-utilization of the full model capacity. ALTER addresses this by introducing dynamism at the timestep level.

### 2.2 Mixture-of-Experts

Mixture-of-Experts (MoE) models [55, 56] conditionally activate only a subset of parameters during inference, offering a promising way to scale computation-expensive neural networks. While extensively utilized in large language models [57, 58], the application of MoE principles to enhance the efficiency of diffusion models remain less explored [59, 60]. APTP [18] highlights the potential of MoE interpretation in diffusion models by using a prompt router to convert each pruned model as an expert specializing in the assigned prompt. However, its reliance on routing each prompt to a fixed expert sub-model throughout the entire diffusion loop restricts model utilization and introduces parameter redundancy due to maintaining multiple expert copies. In light of this, we propose ALTER, where each expert is specialized for a specific range of timesteps and dynamically selected based on timestep embedding routing. Our method uniquely combines timestep-aware routing and structural pruning into a single-stage optimization framework for diffusion acceleration.

## 3 Method

ALTER restructures a standard diffusion UNet into a dynamic ensemble of temporal experts, each defined by a distinct layer-wise pruning configuration applied to the shared backbone. As shown in Fig. 2, a hypernetwork generates these binary pruning masks, and a lightweight router assigns each denoising timestep to an appropriate expert. All components are jointly optimized in a single-stage training process. During inference, the router dynamically activates expert-specific sub-networks conditioned on the timestep.

### 3.1 Preliminaries

**Diffusion Process** Diffusion models [2, 12] capture the data distribution by learning to reverse a predefined noising process applied to the input data. The forward process of the standard diffusion gradually adds Gaussian Noise to the clean input $x_0$ over $T$ timesteps, generating a sequence of increasingly noisy latents $\{x_t\}_{t=1}^{T}$ with the assumption that $x_T$ approximates pure noise. The reverse

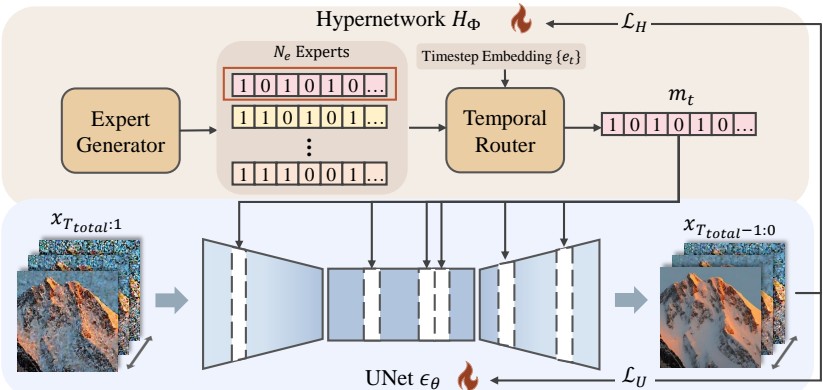

Figure 2: Overview of the ALTER framework. ALTER is a temporal-adaptive-pruning framework for diffusion models, where a hypernetwork generates layer-wise pruning configurations for expert subnetworks and assigns each denoising timestep to a corresponding expert.

process is modeled by a neural network $\epsilon_\theta(x_t, t, c)$ conditioned on the noisy sample $x_t$, the current timestep $t$, and optional context $c$ (e.g., text embeddings). The network is trained denoise $x_t$ into $x_{t-1}$ by predicting the added noise at the $t$-th timestep, following the training objective:

$$\mathcal{L}_{\text{denoise}} = \mathbb{E}_{x_0, \epsilon \sim \mathcal{N}(0,I), t, c} \left[ ||\epsilon - \epsilon_\theta(x_t, t, c)||_2^2 \right], \tag{1}$$

where $x_t = \sqrt{\bar{\alpha}_t} x_0 + \sqrt{1 - \bar{\alpha}_t} \epsilon$, and $\bar{\alpha}_t$ are predefined noise schedule coefficients. To enable the denoising network to adapt its behavior across different noise levels, the discrete timestep $t$ is converted into a continuous vector embedding $e_t \in \mathbb{R}^{D_{emb}}$, where $D_{emb}$ is the embedding dimension. This is often achieved using sinusoidal positional encodings [61] or learned embeddings. The timestep embedding $e_t$ is then integrated into various layers of the neural network by adding it to the hidden activations within its blocks.

**Layer-wise Pruning for Denoising Network.** The denoising network $\epsilon_\theta$ is commonly a UNet [62, 63], featuring an encoder-decoder structure with skip connections. Layer-wise pruning in the UNet architecture refers to the selective omission of entire pre-defined computational blocks, i.e, residual layers or transformer modules. To formalize the definition of pruning, we represent the configuration of a UNet with $N_L$ prunable layers using a binary mask vector $m \in \{0,1\}^{N_L}$, where $m_i = 0$ indicates that the $i$-th layer is pruned, and $m_i = 1$ indicates that it is retained.

### 3.2 Temporal Expert Construction

ALTER employs a trainable hypernetwork $\boldsymbol{H}_\Phi$ to generate layer-wise pruning masks for a shared UNet backbone. These masks define $N_e$ expert configurations, where each expert is equivalent to a layer-wise-pruned sub-network. $\boldsymbol{H}_\Phi$ consists of two trainable components whose trainable parameters jointly form $\Phi$: (1) an Expert Generator $\boldsymbol{G}$ and (2) a Temporal Router $\boldsymbol{R}$.

**Expert Generator.** To generate $N_e$ expert configurations, we employ Expert Generator $\boldsymbol{G}$ to produce a set of pruning masks $\{m_i\} \in [0,1]^{N_e \times N_L}$, where each $m_i \in [0,1]^{N_L}$ serves as the configuration for the $i$-th expert, $i \in \{1, \ldots, N_e\}$. The generator takes as input a set of frozen embeddings $Z \in \mathbb{R}^{N_e \times N_L \times D_{input}}$, initialized with orthogonal vectors, and processes them through several multi-layer perceptrons (MLPs) to produce a matrix of expert layer logits $L_{\text{experts}} \in \mathbb{R}^{N_e \times N_L}$. To approximate binomial distributions while retaining differentiability during training, we apply the Gumbel-Sigmoid function [64] combined with the Straight-Through Estimator (ST-GS) as the final layer on $L_{\text{experts}}$ to obtain $\{m_i\}$.

**Temporal Router.** To select an appropriate expert for the $t$-th timestep in the diffusion process, we employ a Temporal Router $\boldsymbol{R}$ which shares a similar network structure with the expert configuration generator. The temporal router takes the corresponding pre-defined timestep embeddings $e_t \in \mathbb{R}^{D_{emb}}$ from the target UNet's timestep embedding mechanism as input, and maps it to the routing logits $L_t^{\text{routing}} \in \mathbb{R}^{N_e}$ over the $N_e$ expert candidates. Similar to the expert configuration generator, we apply

the Gumbel-Softmax function with the Straight-Through Estimator to the routing logits to obtain the final categorical selection vector $\{s_t\}$.

By routing based on distinct timestep embeddings, the router enables expert specialization across different phases of the denoising trajectory, allowing the model to adaptively assign computation to the most suitable sub-network at each step.

**Differentiable Simulation of Layer-wise Pruning.** To simulate the effect of timestep-specific layer-wise pruning on the diffusion UNet during training, we modify the forward computation of each prunable layer $l \in \{1, \ldots, N_L\}$ as follows:

$$x_{\text{out}} = (1 - (m_t)_l) \cdot x_{\text{in}} + (m_t)_l \cdot f_l(x_{\text{in}}), \tag{2}$$

where $x_{\text{in}}$ is the input to layer $l$, $f_l(\cdot)$ is the original layer computation, and $(m_t)_l \in [0, 1]$ denotes the pruning decision for layer $l$ at timestep $t$. When $(m_t)_l = 0$, the layer is effectively skipped; when $(m_t)_l = 1$, it remains fully active. This formulation allows faithful simulation of pruning behavior while preserving gradient flow, enabling end-to-end optimization. The use of the Straight-Through Estimator (STE) in conjunction with Gumbel-Sigmoid and Gumbel-Softmax ensures differentiable sampling for both the pruning masks and the routing selection vector. During inference, we apply hard pruning by performing actual layer skipping whenever $(m_t)_l = 0$, thereby realizing computational efficiency through expert-specific subnetwork execution.

### 3.3 Optimization Strategy

ALTER, consisting of the shared UNet $\epsilon_\theta$ with parameters $\theta$ and the hypernetwork $\boldsymbol{H}_\Phi$ with parameters $\Phi$, is jointly optimized via a single-stage alternating training scheme. As detailed in Algorithm 1, each training step alternates between the following two optimization phases:

**Shared UNet Backbone Optimization** Given the $N_e$ expert configurations $\mathcal{M}$ generated by the current hypernetwork $\boldsymbol{H}_\Phi$, the goal of shared UNet optimization is to ensure each expert substructure achieves strong denoising performance on its assigned group of timesteps. Thus, the primary training objective is the standard denoising loss $\mathcal{L}_{\text{denoise}}$, computed over masked subnetworks that simulate the active expert at each diffusion timestep. To further improve performance and stabilize training under layer-wise pruning, we optionally employ knowledge distillation from the frozen pretrained teacher model $\epsilon_T$. Specifically, we introduce an output-level distillation loss $\mathcal{L}_{\text{outKD}}$, which encourages the student UNet $\epsilon_S \equiv \epsilon_\theta$ to produce similar denoising outputs as the teacher, and a feature-level distillation loss $\mathcal{L}_{\text{featKD}}$, which aligns intermediate representations between teacher and student. These auxiliary losses are defined as:

$$\mathcal{L}_{\text{outKD}} = \mathbb{E}[\|\epsilon_T(x_t, t, c) - \epsilon_S(x_t, t, c)\|_2^2], \quad \mathcal{L}_{\text{featKD}} = \mathbb{E}[\Sigma_k \|f_T^k(x_t, t, c) - f_S^k(x_t, t, c)\|_2^2], \tag{3}$$

where $f_T^k$ and $f_S^k$ denote the $k$-th block feature activations from the teacher and student models, respectively. The total UNet loss combines the denoising objective and the distillation terms:

$$\mathcal{L}_U = \lambda_{\text{denoise}}\mathcal{L}_{\text{denoise}} + \lambda_{\text{outKD}}\mathcal{L}_{\text{outKD}} + \lambda_{\text{featKD}}\mathcal{L}_{\text{featKD}}, \tag{4}$$

where $\lambda_{\text{outKD}}$ and $\lambda_{\text{featKD}}$ are hyperparameters that control the influence of each distillation term.

**Hypernetwork Optimization.** Based on the current shared UNet backbone, the updated hypernetwork must satisfy three principles: (1) *Performance Preservation*: the temporal expert selected for a given denoising timestep should preserve the denoising capability depicted by the performance loss $\mathcal{L}_{\text{perf}}$ mirrors the UNet objective $\mathcal{L}_U$. It is evaluated using the masks $m_t'$ from the trainable $\boldsymbol{H}_\Phi$ applied to the frozen UNet, ensuring $\boldsymbol{H}_\Phi$ generates effective masks maintaining good generation performance.

(2) *Structure Sparsity*: the temporal sparsity specialization throughout the diffusion trajectory should approach the user-defined computational reduction target. A sparsity regularization loss $\mathcal{L}_{\text{ratio}}(m_t')$ guides $\boldsymbol{H}_\Phi$ to achieve a target overall pruning ratio $p$. This term utilizes a log-ratio matching loss to penalize deviations of the current effective sparsity $S(m_t')$ from $p$. The precise method for calculating $S(m_t')$ based on layer costs and mask activations is detailed in Appendix B.2. $S(m_t')$ is a measure of the active portion of the network's FLOPs under mask $m_t'$. The loss is defined as:

$$\mathcal{L}_{\text{ratio}}(m_t') = \log\left(\frac{\max(S(m_t'), p)}{\min(S(m_t'), p) + \epsilon}\right), \tag{5}$$

**Algorithm 1** ALTER Training Algorithm
___

1: **Input:** Training dataset $\mathcal{D}_{\text{train}}$; UNet $\epsilon_\theta$; Hypernetwork $\boldsymbol{H}_\Phi$; Total training steps $T$; Hypernetwork training end step $T_{\text{end}}$; Loss coefficients $\lambda_{\text{ratio}}$, $\lambda_{\text{balance}}$, $\lambda_{\text{outKD}}$, $\lambda_{\text{featKD}}$.

2: **Initialization:** Initialize UNet parameters $\theta$; Initialize Hypernetwork parameters $\Phi$ (e.g., to output all-active masks).

3: $\mathcal{M} \leftarrow \boldsymbol{H}_\Phi(\text{eval=True})$          ▷ Initial mask configuration from $\boldsymbol{H}_\Phi$

4: **for** $t = 1$ **to** $T$ **do**

5:      Sample a mini-batch $s_b$ from $\mathcal{D}_{train}$;

6:      **if** $t \leq T_{\text{end}}$ **then**          ▷ Hypernetwork Update

7:          $m'_t \leftarrow \boldsymbol{H}_\Phi(s_b, \text{train=True})$          ▷ Differentiable masks for batch $s_b$

8:          $\mathcal{L}_{\text{perf}} \leftarrow \mathcal{L}_{\text{denoise}}(\epsilon_\theta(s_b, \{m'_t\})) + \lambda_{\text{outKD}}\mathcal{L}_{\text{outKD}} + \lambda_{\text{featKD}}\mathcal{L}_{\text{featKD}}$      ▷ $\theta$ frozen

9:          Compute $\mathcal{L}_{\text{ratio}}(m'_t)$ and $\mathcal{L}_{\text{balance}}(m'_t)$;

10:          $\mathcal{L}_{\text{H}} \leftarrow \mathcal{L}_{\text{perf}} + \lambda_{\text{ratio}}\mathcal{L}_{\text{ratio}} + \lambda_{\text{balance}}\mathcal{L}_{\text{balance}}$;

11:          Update $\Phi$ with $\nabla_\Phi \mathcal{L}_{\text{H}}$;

12:          $\mathcal{M} \leftarrow \boldsymbol{H}_\Phi(\text{eval=True})$          ▷ Update mask from new $\boldsymbol{H}_\Phi$

13:      **end if**

14:                                         ▷ UNet Update

15:      Select masks $\{m_t\}$ for $s_b$ from $\mathcal{M}$;

16:      $\mathcal{L}_{\text{UNet}} \leftarrow \mathcal{L}_{\text{denoise}}(\epsilon_\theta(s_b, \{m_t\})) + \lambda_{\text{outKD}}\mathcal{L}_{\text{outKD}} + \lambda_{\text{featKD}}\mathcal{L}_{\text{featKD}}$      ▷ $\Phi$ frozen

17:      Update $\theta$ with $\nabla_\theta \mathcal{L}_{\text{UNet}}$;

18: **end for**

19: **Output:** A fine-tuned UNet $\epsilon_\theta$ and temporal mask decisions $\{m_t\}$.
___

where $\epsilon$ is a small constant for numerical stability.

(3) *Expert diversity*: The router should promote diverse expert selection across timesteps to prevent mode collapse. A router balance loss $\mathcal{L}_{\text{balance}}$ encourages diverse utilization of $N_e$ experts [65, 66]:

$$\mathcal{L}_{\text{balance}} = N_e \sum_{i=1}^{N_e} F_i P_i, \tag{6}$$

where $F_i = \frac{1}{|\mathcal{B}|} \sum_{s_b \in \mathcal{B}} \mathbb{I}(\text{argmax}_j (L_{s_b}^{\text{routing}})_j = i)$ is the fraction of samples in batch $\mathcal{B}$ assigned to expert $i$, and $P_i = \frac{1}{|\mathcal{B}|} \sum_{s_b \in \mathcal{B}} (\text{Softmax}(L_{s_b}^{\text{routing}}))_i$ is the average router probability for expert $i$.

The total hypernetwork objective is

$$\mathcal{L}_H = \mathcal{L}_{\text{perf}} + \lambda_{\text{ratio}}\mathcal{L}_{\text{ratio}} + \lambda_{\text{balance}}\mathcal{L}_{\text{balance}}, \tag{7}$$

where $\lambda_{\text{ratio}}$ and $\lambda_{\text{balance}}$ are scalar coefficients. The differentiability of $m'_t$ is achieved by ST-GS and ST-GSmax, whose implementation details are discussed in Appendix B.3. After updating the hypernet, the mask configuration $\mathcal{M}_U$ for subsequent UNet training steps is refreshed using the updated $\boldsymbol{H}_\Phi$. This alternating optimization strategy allows the UNet to adapt to the dynamic architectures from $\boldsymbol{H}_\Phi$, while $\boldsymbol{H}_\Phi$ learns to generate efficient and effective configurations. The hypernetwork would be trained until $T_{\text{end}}$ is reached.

Such alternating training strategy enables the co-adaptation within ALTER: the UNet progressively recovers its denoising performance under various configurations of layer-wise sparsity, while the hypernetwork keeps refining the configurations and routing of specialized expert configurations tailored to the diffusion process.

## 4 Experiment

### 4.1 Experimental Setup

**Implementation Details.** We experiment on official pretrained diffusion model SDv2.1 [4] to demonstrate the effectiveness of our method. For training, we utilize a randomly sampled 0.3M subset of the LAION-Aestheics V2 (6.5+) [67]. We prune the models as 65% and 10 temporal experts with the loss weight $\lambda_{ratio} = 5$. The bi-level optimization takes 32k steps with global batch size 64, where the hypernetwork is trained for 2 epochs. All experiments are conducted on 2 A100 GPUs.

Table 1: Comparison with BK-SDM and APTP on CC3M and MS-COCO for SD-v2.1. The images are generated at the default resolution 768 and then downsampled to 256.

| Method | Steps | CC3M | | | | | MS-COCO | | | | |
|---|---|---|---|---|---|---|---|---|---|---|---|
| | | MACs (G)($\downarrow$) | Latency (s)($\downarrow$) | FID ($\downarrow$) | CLIP ($\uparrow$) | CMMD ($\downarrow$) | MACs (G)($\downarrow$) | Latency (s)($\downarrow$) | FID ($\downarrow$) | CLIP ($\uparrow$) | CMMD ($\downarrow$) |
| SDv2.1 [4] | 25 | 1384.2 | 4.0 | 17.08 | 30.96 | 0.361 | 1384.2 | 4.0 | 14.29 | 32.17 | 0.537 |
| SDv2.1 [4] | 20 | 1384.2 | 3.2 | 17.37 | 30.89 | 0.374 | 1384.2 | 3.2 | 14.46 | 32.08 | 0.532 |
| BK-SDM-v2 (Base) [53] | 25 | 876.5 | 2.5 | 17.53 | 29.77 | 0.486 | 876.5 | 2.5 | 19.06 | 30.67 | 0.654 |
| BK-SDM-v2 (Small) [53] | 25 | 863.2 | 2.5 | 18.78 | 29.89 | 0.453 | 863.2 | 2.5 | 20.58 | 30.75 | 0.645 |
| APTP (0.85) [18] | 25 | 1182.8 | 3.4 | 36.77 | 30.84 | 0.675 | 1076.6 | 3.1 | 22.60 | 31.32 | 0.569 |
| APTP (0.66) [18] | 25 | 916.3 | 2.6 | 60.04 | 28.64 | 1.094 | 890.0 | 2.5 | 39.12 | 29.98 | 0.867 |
| ALTER (0.65) (Ours) | 25 | 899.7 | 2.6 | **16.74** | **30.94** | 0.393 | 899.7 | 2.6 | **13.70** | 32.17 | 0.536 |
| ALTER (0.65) (Ours) | 20 | 899.7 | 2.1 | 17.14 | 30.92 | **0.392** | 899.7 | 2.1 | 13.89 | **32.18** | **0.533** |
| ALTER (0.60) (Ours) | 20 | 830.5 | 1.9 | 17.87 | 30.88 | 0.479 | 830.5 | 1.9 | 14.24 | 32.17 | 0.596 |

**Datasets.** We evaluate our proposed method on two widely-used diffusion benchmarks: Conceptual Captions 3M [68] (CC3M) and MS-COCO [69]. We evaluate pruning methods using 14k samples in the validation set of CC3M and 30k samples from the MS-COCO 2014's validation split. We sample the images at the default 768 resolution and then resize them to 256 for calculating the metrics with the PNDM [22] sampler following previous works [53, 18]. To compare with previous cache method, we also experiment on the COCO 2017's 5k validation dataset.

**Evaluation Metrics.** We report FID ($\downarrow$) [70] and CMMD ($\downarrow$) [71] to assess image quality, CLIP ($\uparrow$) score [72] to measure text–image alignment, the number of multiply–accumulate operations (MACs) ($\downarrow$) as a proxy for computational cost, and wall-clock inference latency in seconds (Latency) ($\downarrow$). We also report the speedup of ALTER compared with the original model. The MACs and latency are measured with batch size 1 in the A100 GPU platform.

## 4.2 Comparison Results

**Comparison with Static and Sample-wise Dynamic Pruning Method.** We benchmark ALTER against two representative pruning baselines: BK-SDM-v2 [53], which employs static pruning, and APTP [18], which utilizes a sample-wise Mixture-of-Experts (MoE) approach conditioned on input prompts. As detailed in Table 1, ALTER consistently demonstrates superior generative quality and competitive efficiency. Specifically, during a 25-step inference on MS-COCO, ALTER achieves a leading FID of 13.70 and a CLIP score of 32.17, which outperforms BK-SDM-v2 (Small) and APTP (0.66), while operating at comparable computational costs. Similar advantages for ALTER are observed on the CC3M dataset. Remarkably, ALTER's efficiency allows for even stronger performance with fewer steps. With only 20 inference steps, ALTER still surpasses the 25-step performance of both BK-SDM-v2 and APTP, highlighting its superior trade-off between speed and quality. Furthermore, ALTER's performance at both 25 and even 20 steps is not only competitive with but even exceeds that of the original unpruned SDv2.1 model. These comprehensive results confirm that ALTER's strategy of routing masks by timestep, rather than relying on a single global pruning or solely on prompt-based cues, allocates model capacity more precisely to the varying demands of the diffusion process. Qualitative comparisons presented in Figure 3 visually affirm these findings, showcasing that images generated by ALTER exhibit a quality comparable to the unpruned SD-v2.1 model and display fewer artifacts than those produced by BK-SDM-v2.

**Comparison with Cache-Based Method.** We further evaluate ALTER against DiP-GO [30], a leading cache-based acceleration technique, on the MS-COCO 2017 validation set, with results detailed in Table 2. Cache-based methods primarily aim to reduce computation by reusing features from previous timesteps. While effective, they often operate on the original model architecture and may require careful tuning of parameters like cache rate [16] for optimal performance with different inference schedules. In contrast, ALTER's temporal pruning-based approach results in an inference schedule-agnostic model. Once the hypernetwork has determined the temporal expert structures and routing, the resulting dynamically pruned UNet can be deployed with any number of denoising steps without retraining or specific adjustments, offering significant operational flexibility.

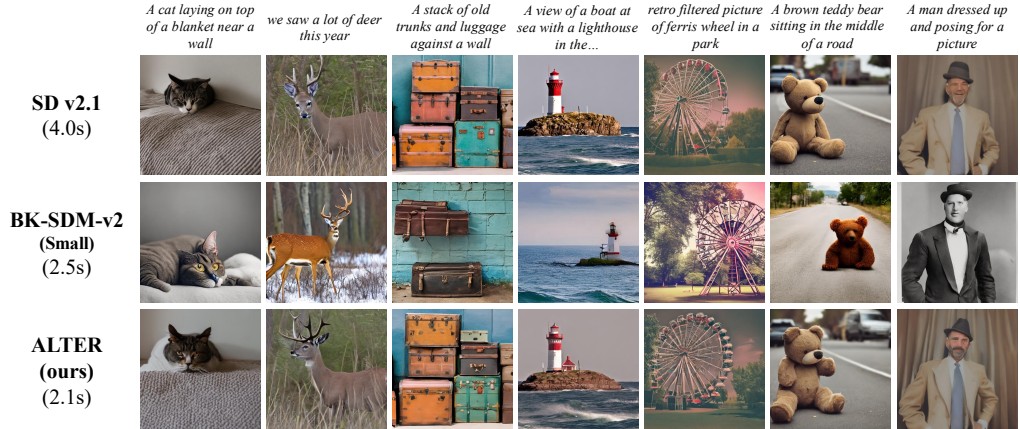

Figure 3: A qualitative comparison with original SDv2.1 and BK-SDM-Small. SDv2.1 and BK-SDM-Small adopt the 25-step PNDM while our method adopts the 20-step inference.

Table 2: Comparison with DiP-GO on the MS-COCO 2017 validation set on SD-2.1. MACs here is the total MACs for all steps.

| Method | MACs($\downarrow$) | Speedup($\uparrow$) | CLIP ($\uparrow$) | FID-5K ($\downarrow$) |
|---|---|---|---|---|
| SDv2.1 (50 steps) [4] | 38.04T | 1.00$\times$ | 31.55 | 27.29 |
| SDv2.1 (20 steps) [4] | 15.21T | 2.49$\times$ | 31.53 | 27.83 |
| DiP-GO (0.7) [30] | 11.42T | 3.02$\times$ | 31.50 | 25.98 |
| ALTER (0.65) (Ours) (20steps) | 9.89T | 3.64$\times$ | **31.62** | **25.25** |
| SDv2.1 (15 steps) | 11.41T | 3.23$\times$ | 30.99 | 27.46 |
| DiP-GO (0.8) [30] | 7.61T | 3.81$\times$ | 30.92 | 27.69 |
| ALTER (0.65) (Ours) (15steps) | 7.42T | 4.37$\times$ | **31.19** | **26.58** |

For a 20-step inference process, ALTER significantly reduces total MACs to 9.89T, which is a substantial improvement over the original SDv2.1 and achieves a 3.64$\times$ speedup compared to the 50-step SDv2.1 baseline, while delivering generative quality that is comparable or even superior (e.g., FID-5K of 25.25 for ALTER vs. 27.83 for SDv2.1 20-steps and 27.29 for SDv2.1 50-steps). ALTER (20 steps) is also more computationally efficient than DiP-GO (0.7) and demonstrates superior performance scores. When the inference step is further reduced to 15 steps, ALTER maintains strong generative quality and achieves better performance compared with the 15-step SDv2.1. It again outperforms DiP-GO (0.8) in both efficiency and generative quality. This robust ability to adapt to various inference steps without re-tuning demonstrates ALTER's suitability for real-time applications.

### 4.3 Abalation Study

**Ablation Study on ALTER's Key Components.** To evaluate the contributions of ALTER's key design components, the usage of multiple specialized experts, the temporal router and the single-stage joint optimization strategy, we conduct a comprehensive ablation study. As shown in Table 3, we compare the full ALTER framework against three variants: (1) "Static", which employs a static global pruning pattern across all timesteps; (2) "Manual", which introduces multiple experts but assigns them to predefined fixed timestep intervals instead of employing a learnable router; (3) "TwoStage", which incorporates ALTER's architecture but adopts a two-stage optimization approach where the hypernetwork is trained first, followed by fine-tuning the UNet.

The results clearly demonstrate the benefits of each component. The "Static" variant, lacking specialized temporal handling, expectedly exhibits the worst performance. Transitioning to the "Manual" approach, which introduces multiple experts for different fixed temporal intervals, yields a significant improvement (e.g., FID drops from 19.03 to 17.91 on CC3M). This demonstrates the importance of having specialized experts for different phases of the denoising process. Further introducing the learnable timestep router in ALTER provides an additional performance boost over

Table 3: Ablation study on pruning and optimization strategies.

| Variant | Multi Expert | Timestep Router | Joint Training | CC3M | | | MS-COCO | | |
|---|---|---|---|---|---|---|---|---|---|
| | | | | FID ($\downarrow$) | CLIP ($\uparrow$) | CMMD ($\downarrow$) | FID ($\downarrow$) | CLIP ($\uparrow$) | CMMD ($\downarrow$) |
| Static | ✗ | ✗ | ✓ | 19.03 | 30.75 | 0.397 | 15.35 | 32.01 | 0.544 |
| Manual | ✓ | ✗ | ✓ | 17.91 | 30.82 | 0.427 | 14.13 | 32.08 | 0.571 |
| TwoStage | ✓ | ✓ | ✗ | 17.65 | 30.88 | 0.401 | 14.45 | 32.02 | 0.539 |
| ALTER | ✓ | ✓ | ✓ | **17.14** | **30.92** | **0.392** | **13.89** | **32.18** | **0.533** |

Table 4: Comparison with TinyFusion and DyDiT for DiT-XL/2 on ImageNet of $256\times256$ resolution.

| Method | Speedup | IS($\uparrow$) | FID($\downarrow$) | sFID($\downarrow$) | Precision($\uparrow$) | Recall($\uparrow$) |
|---|---|---|---|---|---|---|
| DiT-XL/2 [7] (28-layer) | 1.00$\times$ | 277.00 | 2.27 | 4.60 | 0.83 | 0.57 |
| DyDiT-XL [54] ($\lambda$=0.5) | 1.72$\times$ | 248.03 | 2.07 | 4.56 | 0.80 | 0.61 |
| TinyFusion [73] (14-layer) | 1.96$\times$ | 234.50 | 2.86 | 4.75 | 0.82 | 0.55 |
| ALTER (Ours) (avg. 14-layer) | 1.92$\times$ | 254.29 | 2.36 | 4.63 | 0.82 | 0.58 |

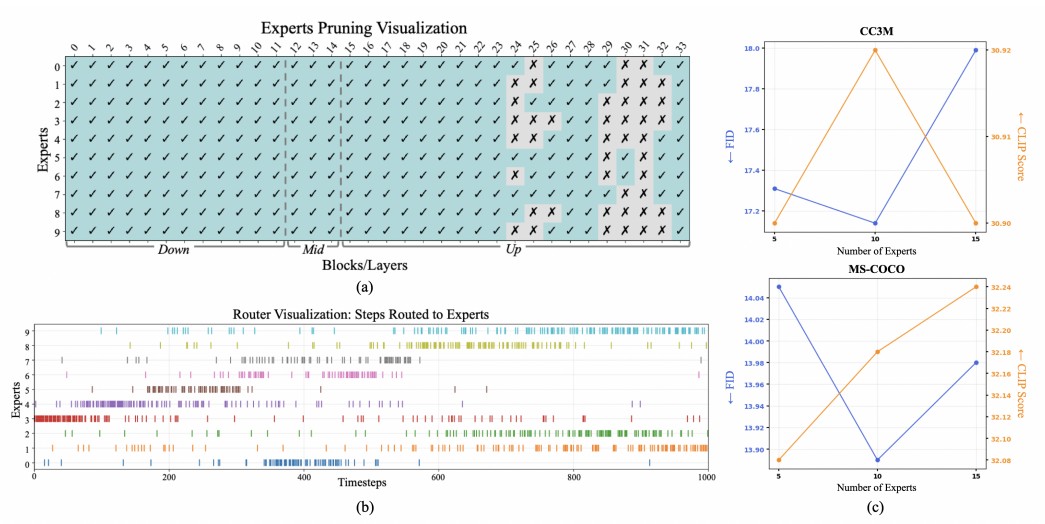

Figure 4: ALTER (0.65)'s temporal experts and router behavior. (a) Visualization of pruning patterns for $N_e = 10$ experts. (b) Visualization of timestep-to-expert routing dynamics. (c) Ablation results for the number of experts on CC3M and MS-COCO 2014.

the "Manual" variant's coarse-grained, fixed assignments, reducing FID to 17.14 on CC3M and 13.89 on MS-COCO, which underscores the benefits of a dynamically learned, fine-grained temporal routing mechanism. Finally, the comparison between ALTER and the "TwoStage" optimization variant shows the critical advantage of our single-stage joint optimization strategy, which facilitates the co-adaptation of UNet's generative ability with the hypernetwork, leading to superior generative quality across all evaluated metrics on both datasets. These ablation experiments collectively validate the effectiveness and contributions of ALTER's key design choices.

**Analysis of Temporal Experts** To understand the characteristics of the temporal experts learned by ALTER, we analyze their pruning patterns, routing dynamics, and the impact of number of experts, as shown in Figure 4. The visualization of pruning patterns for the $N_e = 10$ experts (Fig 4(a)) demonstrates structural diversity. Different experts adopt distinct pruning patterns across the UNet blocks and layers. For instance, some experts being heavily pruned (e.g., experts 3,8,9) while some retain greater capacity (e.g., experts 5-7). This learned structural differentiation allows for temporal specialization. The router dynamics in Fig. 4(b) illustrates how these specialized experts are allocated across different denoising timesteps. We observe that specific experts are predominantly active during certain phases of the diffusion process. Furthermore, the ablation study on the number of experts highlights the importance of an adequate expert number for achieving optimal performance,

as shown in Fig. 4(c). Our chosen configuration of $N_e = 10$ experts consistently yields the best or near-best FID and CLIP scores on both CC3M and MS-COCO datasets. Performance tends to degrade with fewer experts, likely due to insufficient capacity for fine-grained temporal specialization across the entire denoising process. Conversely, increasing to $N_e = 15$ experts does not yield further improvements and can sometimes slightly degrade performance, potentially due to challenges in effectively training a larger set of specialized experts or increased complexity in the routing decisions.

**Generality to DiT.** To assess generality beyond U-Nets, we apply *ALTER* to a DiT-XL/2 backbone [7] and evaluate on ImageNet [74] at $256 \times 256$ using a 250-step DDPM sampler [2]. As shown in Table 4, our method achieves a $1.92\times$ speedup while maintaining an FID score (2.36) comparable to the full model (2.27). With approximately half the layers active on average, our method significantly outperforms static pruning (TinyFusion [73]), especially on key quality metrics like IS (254.29 vs. 234.50) and FID (2.36 vs. 2.86). When comparing to dynamic methods like DyDiT [54], ALTER brings higher speedup. It is important to note that our approaches are orthogonal and complementary. ALTER performs coarse-grained layer-level pruning, while DyDiT performs fine-grained pruning within each block. This empirically confirms that our framework is effective beyond U-Nets and is applicable to modern Transformer-based architectures.

## 5 Conclusion

In this paper, we presented an acceleration framework ALTER that transforms the diffusion model to a mixture of efficient temporal experts by employing a trainable hypernetwork to dynamically generate layer pruning decisions and manage timestep routing to specialized, pruned expert sub-networks throughout the ongoing fine-tuning of the UNet. This timestep-wise approach overcomes the limitations of static pruning and sample-wise dynamic methods, enabling superior model capacity utilization and denoising efficiency. The experiments under various settings and ablation analysis demonstrate the effectiveness of ALTER, which could achieve a $3.64\times$ speedup on the original SDv2.1 with 35% sparsity.

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

# A Limitations

While ALTER employs joint optimization for the co-adaptation of the U-Net and hypernetwork, a final fine-tuning stage for the dynamically pruned U-Net is still found to be a necessary step after $T_{\text{end}}$ is reached. This final tuning phase is crucial for fully recovering any potential performance degradation incurred during the temporal-wise structure pruning and for allowing the U-Net to optimally adapt to its newly defined sparse, dynamic structure, thereby maximizing its generative quality.

# B More Details of ALTER

This appendix provides further details on the architectural components of our hypernetwork $\boldsymbol{H}_\Phi$, the calculation of sparsity metrics, the implementation of Gumbel-Sigmoid and Gumbel-Softmax with straight-through estimation, and the handling of dynamic masks.

## B.1 Detailed Hypernetwork Architectures

The hypernetwork $\boldsymbol{H}_\Phi$ consists of two main modules: the Expert Structure Generator and the Temporal Router. Their architectures are detailed in Table 5 and Table 6. We use $D_{\text{input}}$ for the dimensionality of the learnable embeddings for the Expert Structure Generator, $D_{\text{expert}}$ for its hidden layer, $D_{emb}$ for the UNet's timestep embedding dimension, and $D_{\text{router}}$ for the Temporal Router's hidden layer. Specific values used in our experiments are provided in Appendix C.

Table 5: Architecture of the Expert Structure Generator module in $\boldsymbol{H}_\Phi$. Processes $N_e \times N_L$ input embeddings to produce $N_e \times N_L$ logits.

| Component | Details |
|---|---|
| Input | Frozen Embeddings $Z \in \mathbb{R}^{N_e \times N_L \times D_{\text{input}}}$ (Initialized orthogonally) |
| Hidden Layer | Linear($D_{\text{input}} \to D_{\text{expert}}$) $\to$ LayerNorm($D_{\text{expert}}$) $\to$ ReLU |
| Output Layer | Linear($D_{\text{expert}} \to 1$, no bias) |
| Output | Logits $L_{\text{experts}} \in \mathbb{R}^{N_e \times N_L}$ (Squeeze last dim) |

Table 6: Architecture of the Temporal Router module ($\boldsymbol{R}_{\text{router}}$) in $\boldsymbol{H}_\Phi$. Processes each timestep embedding $e_t$ to produce $N_e$ routing logits.

| Component | Details |
|---|---|
| Input | Timestep Embedding $e_t \in \mathbb{R}^{D_{emb}}$ (from $E_{\text{timesteps}} \in \mathbb{R}^{T_{\text{total}} \times D_{emb}}$) |
| Hidden Layer | Linear($D_{emb} \to D_{\text{router}}$) $\to$ LayerNorm($D_{\text{router}}$) $\to$ ReLU |
| Output Layer | Linear($D_{\text{router}} \to N_e$, no bias) |
| Output | Routing Logits $L_t^{\text{routing}} \in \mathbb{R}^{N_e}$ (for a single $e_t$) 
 (Total output $L^{\text{routing}} \in \mathbb{R}^{T_{\text{total}} \times N_e}$) |

## B.2 Calculation of Current Sparsity

The current effective sparsity $S(m_t')$ used in the sparsity regularization term $\mathcal{L}_{\text{ratio}}$ (Equation 5), quantifies the active computational cost relative to the total potential cost. It is calculated based on the soft (differentiable) pruning masks $m_t'$ generated by $\boldsymbol{H}_\Phi$ during its training phase. Let $(m_t')_l$ be the soft mask value for the $l$-th prunable layer and $c_l$ be the pre-calculated computational cost (e.g., FLOPs) of that layer. The current sparsity $S(m_t')$ is the weighted average of these mask values, normalized by the total cost of all prunable layers:

$$S(m_t') = \frac{\sum_{l=1}^{N_L} (m_t')_l \cdot c_l}{\sum_{l=1}^{N_L} c_l} \tag{8}$$

This formulation ensures that $S(m'_t)$ represents the fraction of the total computational cost that remains active under the current soft mask $m'_t$. The per-layer costs $c_l$ are typically profiled once from the pretrained full UNet architecture.

### B.3 ST-GS & ST-GSmax Implementation Details

To enable differentiable sampling of discrete architectural decisions, we employ ST-GS and ST-GSMax techniques.

**ST-GS Sampling.** For layer pruning decisions within the Expert Structure Generator, logits $(L_{\text{experts}})_{e,l}$ are converted to binary masks. We sample Gumbel noise $g \sim \text{Gumbel}(0,1)$. Given a logit $L$, a sampling temperature $\tau_g = 0.4$, and an offset $b_g = 4$, the soft decision $y_g$ is computed:

$$y_g = \sigma \left( \frac{L + g + b_g}{\tau_g} \right)$$

where $\sigma(\cdot)$ is the sigmoid function. To obtain a hard binary decision $(M_{\text{experts}})_{e,l}$ while maintaining differentiability for training, we use STE [75]: $(M_{\text{experts}})_{e,l} = \text{round}(y_g)$. During backpropagation, the gradient is passed through $y_g$ as if the rounding operation were the identity function for values not exactly at 0.5, effectively using $\nabla(\text{round}(y_g)) \approx \nabla y_g$. This results in a tensor of '0's and '1's for the forward pass, forming the pruning masks.

**ST-GSmax Sampling.** For temporal routing decisions, given a vector of routing logits $L_t^{\text{routing}} \in \mathbb{R}^{N_e}$, a temperature $\tau_r = 0.4$, an offset $b_r = 0$, and a vector of Gumbel noise $\boldsymbol{G} \in \mathbb{R}^{N_e}$ (components $g_j$ sampled independently), the soft Gumbel-Softmax probabilities $y_r \in [0,1]^{N_e}$ are:

$$(y_r)_j = \frac{\exp(((L_t^{\text{routing}})_j + g_j + b_r)/\tau_r)}{\sum_{k=1}^{N_e} \exp(((L_t^{\text{routing}})_k + g_k + b_r)/\tau_r)}$$

To obtain a hard one-hot selection vector $s_t$ for the forward pass while enabling gradient flow for training, STE is applied. Determine the index of the maximally activated expert: $j^* = \text{argmax}_j (y_r)_j$. Create a hard one-hot vector $(s_t^{\text{hard}})_j = \mathbf{1}_{j=j^*}$. The output used in computation, which enables STE, is $s_t = (s_t^{\text{hard}} - y_r).\text{detach}() + y_r$. This makes $s_t$ a one-hot vector in the forward pass, while its gradient in the backward pass is taken with respect to the soft probabilities $y_r$.

### B.4 Dynamic Mask Handling

The dynamic pruning masks, central to ALTER's efficiency, are generated and applied as follows. For each timestep $t$, an effective mask $m_t \in [0,1]^{N_L}$ is composed from the $N_e$ expert structure masks $M_{\text{experts}}$ and the per-timestep routing selection $s_t$. When training the UNet or during inference, deterministic binary masks ($m_t^{\text{hard}}$, forming the set $\mathcal{M}_{\text{UNet}}$) are derived by setting the hypernetwork $\boldsymbol{H}_\Phi$ to evaluation mode. For training $\boldsymbol{H}_\Phi$ itself, differentiable soft versions of these masks ($m'_t$) are utilized.

The UNet architecture has a specific block structure. The generated flat mask vector (whether $m_t^{\text{hard}}$ or $m'_t$) of length $N_L$ is mapped to the corresponding prunable layers within the UNet using a utility function. This ensures that the correct mask component $(m_t)_l$ is applied to the $l$-th prunable layer $f_l$. The application of this component follows Equation 2 from the main text: $x_{\text{out}} = (1 - (m_t)_l) \cdot x_{\text{in}} + (m_t)_l \cdot f_l(x_{\text{in}})$. This general mechanism applies to all designated prunable layers. It is noteworthy that in UNet architectures, particularly in the decoder pathway, layers $f_l$ often process inputs that are a result of concatenating features from a previous layer in the same pathway with features from an earlier encoder layer via a skip connection. Our layer-wise pruning applies directly to such layers $f_l$: if $f_l$ is skipped due to $(m_t)_l \approx 0$, its entire operation, including its specific handling of any skip-connected features, is bypassed.

## C  More Implementation Details

Here we provide additional details regarding our experimental setup, training details and hyperparameter configurations for ALTER. All models are trained or fine-tuned starting from official pretrained

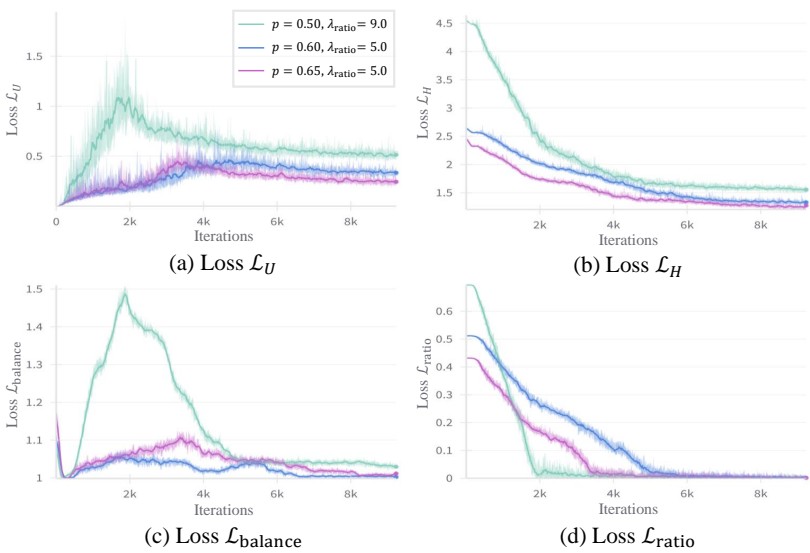

Figure 5: The training dynamics given different ratios $p$ and $\lambda_{\text{ratio}}$ weights.

SD-2.1 [4] checkpoints. The training is conducted at a resolution of $512 \times 512$ pixels using a randomly sampled subset of 0.3M images from the LAION-Aesthetics V2 dataset (rated 6.5+) [67]. The total optimization process spans $T = 32,000$ steps with a global batch size of 64, distributed across 2 A100 GPUs (32 samples per GPU). The hypernetwork $\boldsymbol{H}_\Phi$ is actively trained until $T_{\text{end}}$, which is set to 2 epochs. We employ $N_e = 10$ temporal experts and sparsity ratio $p = 0.65$ for our main experiments. We use the AdamW optimizer [76] for both the U-Net parameters $\theta$ and the hypernetwork parameters $\Phi$. A linear learning rate warm-up is applied for the first 250 iterations for both optimizers. After the warm-up phase, constant learning rates are used: $1 \times 10^{-5}$ for the U-Net and $7 \times 10^{-5}$ for the hypernetwork. For the GS function used in the Expert Generator, the temperature $\tau_g$ is set to 0.4 and the offset $b_g$ to 4.0, which initializes expert masks to all-ones, promoting layer activity at the start of training and preventing an overly detrimental impact on the U-Net from aggressive initial pruning. For the GSmax function in the Temporal Router, the temperature $\tau_r$ is 0.4 and the offset $b_r$ is 0.0. Since the denoising loss $L_{\text{denoise}}$ tends to result in unstable optimization and slow convergence. The loss weights for the objective functions are set to $\lambda_{\text{denoise}} = 1e-4$, $\lambda_{\text{outKD}} = \lambda_{\text{featKD}} = \lambda_{\text{balance}=1.0}$ and $\lambda_{\text{ratio}} = 5.0$. For hypernetwork's implementation, we set $D_{\text{input}} = 64$, $D_{\text{expert}} = 256$, $D_{\text{router}} = 64$, and the $D_{emb}$ is depedent on the pre-trained official SDv2.1 model. For generating samples for evaluation, we use classifier-free guidance (CFG) [77] with the default scale of the respective SDv2.1 checkpoint (e.g., 7.5). Image generation is performed using the PNDM sampler [22], following practices in prior works for consistency.

## D More Experimental Results

### D.1 Training Dynamics

In Fig. 5, we visualize the training dynamics for different pruning ratios $p$ and their associated $\lambda_{\text{ratio}}$, highlighting the co-adaptation between the shared UNet backbone and the hypernetwork. As shown in Fig. 5(a), the UNet loss $\mathcal{L}_U$ generally follows a two-phase pattern: it first increases, especially under high sparsity targets (e.g., $p = 0.50$), and then gradually decreases. This early rise reflects a transient mismatch, as the hypernetwork begins to sample diverse pruning structures across timesteps while the shared UNet parameters have not yet adapted to the corresponding architectural variations. Over time, however, $\mathcal{L}_U$ steadily declines, indicating that the shared UNet progressively learns to meet the performance demands imposed by a wide range of expert sub-structure configurations at different timesteps. The final converged $\mathcal{L}_U$ remains slightly higher under more aggressive pruning configurations, reflecting the expected trade-off between model sparsity and generative fidelity.

Table 7: Comparison of Image Reward and HPSv2 metrics. The images are sampled with 25-step inference.

| Method | MACs(G)($\downarrow$) | Image Reward($\uparrow$) | HPSv2($\uparrow$) |
|---|---|---|---|
| SDv2.1 | 1384.2 | 0.1139 | 24.47 |
| BK-SDM-v2-Base | 876.5 | -0.0365 | 23.97 |
| APTP (0.66) (COCO) | 916.3 | -0.8706 | 19.08 |
| APTP (0.66) (CC3M) | 916.3 | -1.2226 | 18.19 |
| ALTER (0.65) (Ours) | 899.7 | 0.1494 | 24.54 |
| ALTER (0.60) (Ours) | 830.5 | 0.1487 | 24.36 |

Table 8: Comparison of SSIM and PSNR metrics.

| Method | Steps | MACs (G)($\downarrow$) | Latency (s)($\uparrow$) | CC3M | | MS-COCO | |
|---|---|---|---|---|---|---|---|
| | | | | SSIM ($\uparrow$) | PSNR ($\uparrow$) | SSIM ($\uparrow$) | PSNR ($\uparrow$) |
| SDv2.1 | 25 | 1384.2 | 4.0 | - | - | - | - |
| SDv2.1 | 20 | 1384.2 | 3.2 | 0.3529 | 13.71 | 0.1582 | 10.24 |
| BK-SDM-v2 (Base) | 25 | 876.5 | 2.5 | 0.1208 | 9.42 | 0.1551 | 9.47 |
| APTP (0.66) | 25 | 916.3 | 2.6 | 0.0782 | 8.99 | 0.0730 | 8.83 |
| ALTER (0.65) (Ours) | 25 | 899.7 | 2.6 | 0.1233 | 9.90 | 0.3261 | 12.81 |
| ALTER (0.65) (Ours) | 20 | 899.7 | 2.1 | 0.1247 | 9.95 | 0.2484 | 11.61 |
| ALTER (0.60) (Ours) | 20 | 830.5 | 1.9 | 0.1224 | 9.94 | 0.1898 | 10.73 |

In contrast, the hypernetwork's loss $\mathcal{L}_H$ (Fig. 5(b)) exhibits a steady and consistent decline across all configurations, indicating that $\boldsymbol{H}_\Phi$ is effectively optimizing its composite objective, which includes the UNet performance term $\mathcal{L}_{\text{perf}}$ and the associated regularization components. This consistent improvement suggests that the hypernetwork progressively discovers better expert structures and routing strategies alongside the UNet's parameter update, while gradually approaching the desired computational reduction target.

The regularization losses further elucidate this process. The sparsity loss $\mathcal{L}_{\text{ratio}}$ (Fig. 5(d)) decreases smoothly over time, confirming that the hypernetwork successfully aligns the average computational cost of selected experts with the target sparsity level $p$. The convergence rate is influenced by both the sparsity target and the loss weight $\lambda_{\text{ratio}}$. Similarly, the router balance loss $\mathcal{L}_{\text{balance}}$ (Fig. 5(c)) shows a downward trend, indicating that the temporal router gradually learns to distribute computation evenly across experts. Notably, under the most aggressive pruning setting ($p = 0.50$), $\mathcal{L}_{\text{balance}}$ initially spikes, suggesting a greater challenge in achieving balanced routing when sparsity constraints are tighter.

This intricate interplay—where $\mathcal{L}_H$ steadily improves while $\mathcal{L}_U$ undergoes a transient perturbation—highlights the bi-level nature of ALTER's optimization process, which enables the emergence of efficient, dynamically structured expert configurations.

This intricate interplay, where $\mathcal{L}_H$ steadily improves while $\mathcal{L}_U$ overcomes an initial perturbation, highlights the bi-level nature of the optimization problem that ALTER navigates to achieve its efficient, dynamically structured experts with high performance preservation.

### D.2 More Evaluation Metrics

**ImageReward and HPSv2** We benchmark our models using advanced preference metrics, ImageReward [78] and Human Preference Score v2 (HPSv2) [79] in order to provide a more comprehensive evaluation of generation quality. These metrics assess perceptual quality and human aesthetic preference. As shown om Table 7, our method ranks highest on both metrics, even outperforming the original SDv2.1, confirming that our efficiency gains do not come at the cost of visual quality.

**SSIM and PSNR** We provide the SSIM and PSNR to show the deviation of different methods from the original SDv2.1 25-step inference. As shown in Table 8, our method ALTER not only matches the MACs of prior work but also achieves significantly higher SSIM and PSNR scores,

Table 9: Scalability with token reduction method ToMeSD on the MS-COCO 2017 validation set.

| Method | Steps | ToMe r% | Speedup(↑) | CLIP (↑) | FID-5K (↓) |
|---|---|---|---|---|---|
| SDv2.1 | 50 | 0 | 1.00× | 31.55 | 27.29 |
| SDv2.1 | 20 | 0 | 2.49× | 31.53 | 27.83 |
| ALTER (0.65) (Ours) | 20 | 0 | 3.64× | 31.62 | 25.25 |
| | 20 | 40 | 4.93× | 31.48 | 25.93 |
| w/ ToMeSD [80] | 20 | 50 | 5.75× | 31.34 | 26.89 |
| | 20 | 60 | 6.54× | 31.24 | 27.96 |

Table 10: Ablation study on the sparsity ratio $p$. The images are sampled with 20-step inference.

| Method | CC3M | | | | | MS-COCO | | | | |
|---|---|---|---|---|---|---|---|---|---|---|
| | MACs (G)(↓) | Latency (s)(↓) | FID (↓) | CLIP (↑) | CMMD (↓) | MACs (G)(↓) | Latency (s)(↓) | FID (↓) | CLIP (↑) | CMMD (↓) |
| SDv2.1 [4] | 1384.2 | 3.2 | 17.37 | 30.89 | 0.374 | 1384.2 | 3.2 | 14.46 | 32.08 | 0.532 |
| ALTER (0.65) | 899.7 | 2.1 | 17.14 | 30.92 | 0.392 | 899.7 | 2.1 | 13.89 | 32.18 | 0.533 |
| ALTER (0.60) | 830.5 | 1.9 | 17.87 | 30.88 | 0.479 | 830.5 | 1.9 | 14.24 | 32.17 | 0.596 |
| ALTER (0.55) | 761.3 | 1.8 | 18.27 | 30.87 | 0.563 | 761.3 | 1.8 | 14.59 | 32.16 | 0.679 |
| ALTER (0.50) | 692.1 | 1.6 | 18.99 | 30.74 | 0.612 | 692.1 | 1.6 | 14.70 | 32.05 | 0.727 |

which demonstrates that our approach better preserves the generative characteristics of the original model.

### D.3 Generality to DiT

To assess generality beyond U-Nets, we apply *ALTER* to a DiT-XL/2 backbone [7] and evaluate on ImageNet [74] at 256×256 using a 250-step DDPM sampler [2].

As shown in Table 4, our method achieves a 1.92× speedup while maintaining an FID score (2.36) comparable to the full model (2.27). With approximately half the layers active on average, our method significantly outperforms static pruning (TinyFusion [73]), especially on key quality metrics like IS (254.29 vs. 234.50) and FID (2.36 vs. 2.86). When comparing to dynamic methods like DyDiT [54], ALTER brings higher speedup. It is important to note that our approaches are orthogonal and complementary. ALTER performs coarse-grained layer-level pruning, while DyDiT performs fine-grained pruning within each block. This empirically confirms that our framework is effective beyond U-Nets and is applicable to modern Transformer-based architectures.

### D.4 Scalability with Other Accelerators

ALTER's design makes it orthogonal to and compatible with other diffusion acceleration methods such as diffusion cache and token reduction. We conduct an experiment combining ALTER with a token reduction method ToMeSD [80] to demonstrate ALTER's scalability. For this combination, we follow the default setting of ToMeSD. As shown in Table 9, we boost the total speedup from 3.64× to an impressive 6.54× with only a minor, controllable trade-off in performance scores. This empirically validates the excellent scalability of our method.

### D.5 Ablation Study of Sparsity Ratio

We investigate the impact of varying the target sparsity ratio $p$ on ALTER's performance. Results are reported in Table 10 under the unified 20-step inference setting, showing a clear trade-off between sparsity and generation quality. As $p$ decreases from 0.65 to 0.50, performance metrics degrade gradually. Notably, ALTER variants with $p = 0.65$ and $p = 0.60$ match the generative quality of the original SDv2.1 baseline while significantly reducing computational cost. Even the most aggressive setting, ALTER (0.50), maintains competitive performance (e.g., FID of 14.70 and CLIP score of 32.05 on MS-COCO) with the highest efficiency gains. These results highlight ALTER's flexibility and its ability to balance quality and efficiency under different sparsity budgets.

Table 11: Ablation study on different components of $\mathcal{L}_U$ for the "Static" Variant.

| $\mathcal{L}_U$ | CC3M | | | MS-COCO | | |
|---|---|---|---|---|---|---|
| | FID ($\downarrow$) | CLIP ($\uparrow$) | CMMD ($\downarrow$) | FID ($\downarrow$) | CLIP ($\uparrow$) | CMMD ($\downarrow$) |
| $\mathcal{L}_{\text{denoise}}$ | 23.76 | 29.93 | 0.531 | 21.57 | 30.91 | 0.709 |
| $\mathcal{L}_{\text{denoise}} + \mathcal{L}_{\text{outKD}}$ | 20.15 | 30.54 | 0.424 | 15.98 | 31.79 | 0.549 |
| $\mathcal{L}_{\text{denoise}} + \mathcal{L}_{\text{outKD}} + \mathcal{L}_{\text{featKD}}$ | **19.03** | **30.75** | **0.397** | **15.35** | **32.01** | **0.544** |

Table 12: Ablation of post-$T_{\text{end}}$ fine-tuning.

| Method | Steps | CLIP($\uparrow$) | FID-5K($\downarrow$) |
|---|---|---|---|
| SDv2.1 | 20 | 31.53 | 27.83 |
| ALTER (0.65) ($T_{\text{end}}$) | 20 | 31.54 | 25.98 |
| ALTER (0.65) ($T_{\text{total}}$) | 20 | 31.62 | 25.25 |
| ALTER (0.65) ($T_{\text{end}}$) | 15 | 31.05 | 27.16 |
| ALTER (0.65) ($T_{\text{total}}$) | 15 | 31.19 | 26.58 |

### D.6 Ablation on Loss Components of $\mathcal{L}_U$

We investigate the impact of incorporating knowledge distillation losses into the UNet's training objective $\mathcal{L}_U$ for the "Static" variant. As shown in Table 11, introducing the distillation of the output and block features are able to enhance the performance of the statically pruned model, which guides our choice of loss components.

### D.7 Ablation on Final Finetuning

As the results in Table 12 demonstrate, the model's performance after $T_{\text{end}}$ is already very strong, achieving the majority of the final quality. The final fine-tuning provides a consistent and important boost across both metrics and step counts, further improving both FID and CLIP scores.

### D.8 More Qualitative Results

We provide more qualitative results compared with the 25-step original SDv2.1 and BK-SDM-V2-Small. As shown in Fig. 6, ALTER exhibits a quality comparable to the unpruned SD-v2.1 model and displays fewer artifacts than those produced by BK-SDM-v2.

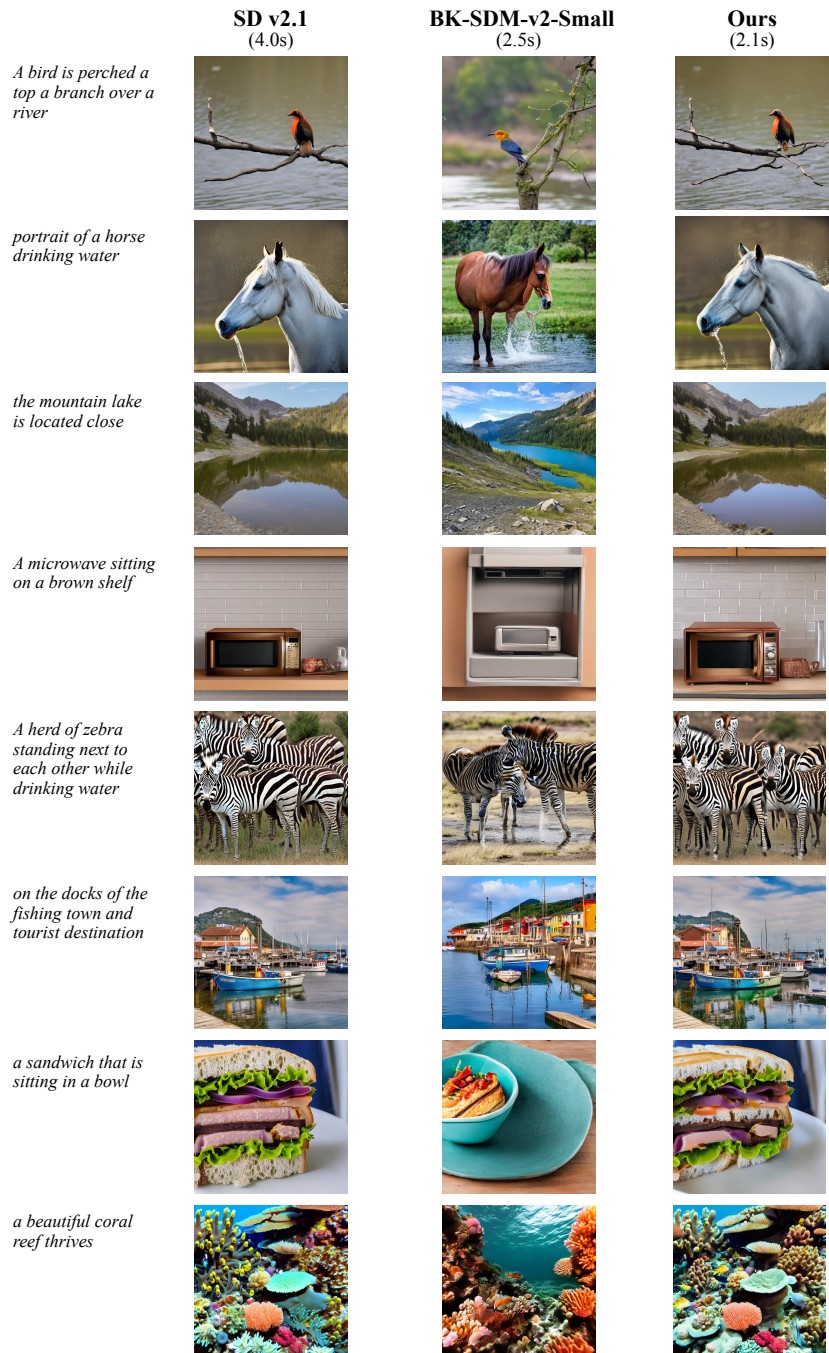

Figure 6: More qualitative results compared with original SDv2.1 and BK-SDM-Small. SDv2.1 and BK-SDM-Small adopt the 25-step PNDM while our method adopts the 20-step inference.

