# OpenReview forum: "ALTER: All-in-One Layer Pruning and Temporal Expert Routing for Efficient Diffusion Generation"
_NeurIPS.cc/2025/Conference — NeurIPS 2025 poster_

### Official Review · Reviewer_C2Gm · 2025-06-05

**Clarity:** 3
**Significance:** 3
**Originality:** 3
**Rating:** 4
**Confidence:** 5

**Summary:**

This paper proposes to view layer-pruned sub-networks as experts, and learns to dynamically select different sub-networks for different diffusion timesteps. A 2D binary mask is used to represent the layer selection results for each expert, and a router selects an expert based on the timestep. The method effectively reduce the average computation of SD2.1 with comparable generation quality.

**Questions:**

1. It can be observed from Fig. 4 that each expert's sparsity level is not high (most layers are still preserved). I'm curious why the speedup with 25 steps is still significant in Tab. 1.

2. In T2I settings, GenEval [3] might be helpful to evaluate the generation ability more comprehensively. For SD3 or FLUX-based models, human evaluation is also recommended.

[3] GenEval: An Object-Focused Framework for Evaluating Text-to-Image Alignment.

**Ethical Concerns:**

["NO or VERY MINOR ethics concerns only"]

**Final Justification:**

After reading the rebuttal and other reviewers' comments, I tend to maintain my score

**Limitations:**

Yes

**Quality:**

3

**Strengths And Weaknesses:**

### Strengths
1. The motivation is clear and reasonable;

2. The method is novel and sound;

3. The results on SD2.1 are solid, and the realistic speedup is significant.

### Weaknesses

Overall, I think it's an interesting work. My major concern is the insufficient experimental validation.

1. The validated model is limited on SD2.1. As we all know, SD2.1 is a UNet structure, and the currently popular structures like SD3, Flux, Hunyuan and Wanx are based on DiT architectures. It would be more convincing if the authors validate the method on more DiT architectures, including the original DiT and the T2X models (FLUX and Wanx).

2. The compared baselines are insufficient.
    - Static layer-pruning method: FLUX-lite (https://huggingface.co/Freepik/flux.1-lite-8B).
    - Dynamic methods such as DyDiT [1] and TPDM [2].
    - Few-step methods such as SD-turbo (based on SD2.1, https://huggingface.co/stabilityai/sd-turbo) and FLUX-turbo (https://huggingface.co/alimama-creative/FLUX.1-Turbo-Alpha).


[1] DyDiT++: Dynamic Diffusion Transformers for Efficient Visual Generation

[2] Schedule On the Fly: Diffusion Time Prediction for Faster and Better Image Generation

---

> ### Author Rebuttal · Authors · 2025-07-31
>
> Thank you for your review and for recognizing the novelty and solid performance of our work. We have performed several new experiments to address your primary concerns.
>
> ---
>
> ## W1 - Generalization to DiT Architectures
>
> We agree that demonstrating generality is crucial. To address this, we have **conducted a new experiment applying ALTER to a DiT-XL/2 backbone**. The results show that our method achieves a ~1.92x speedup while maintaining an FID score (2.36) comparable to the full model (2.27).
>
> With approximately half the layers active on average, **our method significantly outperforms static pruning (TinyFusion), especially on key quality metrics like IS (254.29 vs. 234.50) and FID (2.36 vs. 2.86).** When comparing to dynamic methods like DyDiT, ALTER brings higher speedup. It's important to note that our approaches are **orthogonal and complementary**. ALTER performs coarse-grained **layer-level pruning**, while DyDiT performs fine-grained pruning _within_ each block. These new results demonstrate that ALTER is a general and effective framework for accelerating various diffusion architectures.
>
> | Model                        | Speedup      | IS ($\uparrow$) | FID ($\downarrow$) | sFID $(\downarrow)$ | Precision ($\uparrow$) | Recall ($\uparrow$) |
> | ---------------------------- | ------------ | --------------- | ------------------ | ------------------- | ---------------------- | ------------------- |
> | DiT-XL/2 [4] (28-layer)      | 1.00$\times$ | 277.00          | 2.27               | 4.60                | 0.83                   | 0.57                |
> | DyDiT-XL [5] ($\lambda=0.5$) | 1.72$\times$ | 248.03          | 2.07               | 4.56                | 0.80                   | 0.61                |
> | TinyFusion [6] (14-layer)    | 1.96$\times$ | 234.50          | 2.86               | 4.75                | 0.82                   | 0.55                |
> | ALTER (Ours) (avg. 14-layer) | 1.92$\times$ | 254.29          | 2.36               | 4.63                | 0.82                   | 0.58                |
>
> We have not implemented ALTER on SD3 and FLUX because of the limited computational resources and rebuttal time, but our successful application to DiT provides strong evidence that our approach is a general framework for accelerating various diffusion backbones.
>
> ---
>
> ## W2 - Comparison with Other Baselines
>
> Thank you for providing this comprehensive list. Here is how we position our work with respect to these methods:
> - **DyDiT++ [4] & TPDM [5] :** As shown above, our DiT experiment now includes a comparison with DyDiT. But it's important to note that our approaches are **orthogonal and complementary**. ALTER performs coarse-grained **layer-level pruning**, while DyDiT performs fine-grained pruning _within_ each block. TPDM is an orthogonal approach that accelerates inference by predicting and skipping entire timesteps. Our method, which prunes layers _within_ a step, could be combined with TPDM to accelerate the remaining, non-skipped steps.
>
> - **Few-step & Distilled Models (SD-turbo, FLUX-turbo):** These methods represent a different family of acceleration that uses distillation to train a specialized model for extremely few-step (e.g., 4 step) inference. Our work focuses on accelerating the original, pre-trained model for general-purpose use across various step counts (e.g., 20-50 steps) without requiring a full distillation process. Besides, step distillation methods are computationally intensive, often demanding thousands of GPU hours, whileour ALTER method is lightweight, requiring **only about 48 A100 hours** to fine-tune the SDv2.1 model.
>
> - **FLUX-lite:** This is a statically pruned version of the FLUX architecture. Our primary experiments focus on accelerating the public SD2.1 and DiT backbones. Where we do include a statically pruned version of the SDv2.1 (BK-SDM-v2) and the DiT-XL/2 (TinyFusion).
>
> We have **updated our related work section** to include a detailed discussion of these methods, clarifying the different axes of acceleration and contextualizing our contribution.
>
> ---
>
> ## Q1 - Why the speedup with 25 steps is still significant in Tab. 1.
>
> As shown in Fig.4, the pruned layers are almost at the last part, which is at high-resolution considering the UNet's architecture. The significant speedup arises from the **highly non-uniform computational cost of layers in a U-Net**. Although the pruned layer's number is not big, prunning the high-resolution could benefit more for acceleration compared with the middle layers.
>
> ---
>
> ## Q2 - GenEval [6] and Human Evaluation
>
> Thank you for the valuable suggestion to use GenEval. We made a significant effort to incorporate it but encountered unresolved technical issues with the official GenEval repository's environment setup, which we were unable to fix within the limited rebuttal period. And these technical problems are very common as shown in the github repository's opening issues. Thanks for your understanding.
>
> However, to follow the spirit of your suggestion for a more comprehensive evaluation we have benchmarked our method using **ImageReward** [7] and Human Preference Score v2 (**HPSv2**) [8], two state-of-the-art models designed to measure human aesthetic and semantic preferences. We calculate these two metrics on their own benchmark datasets. Since APTP is prompt-based pruning, we use the prompt split method they provided to divide the benchmark's prompts to different experts finetuned on MS-COCO and CC3M.
>
> As shown in the following two tables, the results are highly encouraging. On ImageReward, **our ALTER models rank #1 (0.65 MACs) and #2 (0.60 MACs)**, outperforming prior methods and even the original SDv2.1 baseline, which indicates a higher alignment with human preference. Similarly, on the fine-grained HPSv2 benchmark, our method consistently achieves the highest average score, demonstrating robust performance across diverse visual styles from animation to photo-realism.
>
> | Model               | MACs (G) | Image Reward ($\uparrow$) | Rank |
> | ------------------- | -------- | ------------------------- | ---- |
> | SDv2.1              | 1384.2   | 0.1139                    | 3    |
> | BK-SDM-v2 (Base)    | 876.5    | -0.0365                   | 4    |
> | APTP (0.66) (COCO)  | 916.3    | -0.8706                   | 5    |
> | APTP (0.66) (CC3M)  | 916.3    | -1.2226                   | 6    |
> | ALTER (0.65) (Ours) | 899.7    | 0.1494                    | 1    |
> | ALTER (0.60) (Ours) | 830.5    | 0.1487                    | 2    |
>
> | Model               | MACs (G) | HPSv2 ($\uparrow$) | Rank |
> | ------------------- | -------- | ------------------ | ---- |
> | SDv2.1              | 1384.2   | 24.47              | 2    |
> | BK-SDM-v2 (Base)    | 876.5    | 23.97              | 4    |
> | APTP (0.66) (COCO)  | 916.3    | 19.08              | 5    |
> | APTP (0.66) (CC3M)  | 916.3    | 18.19              | 6    |
> | ALTER (0.65) (Ours) | 899.7    | 24.54              | 1    |
> | ALTER (0.60) (Ours) | 830.5    | 24.36              | 3    |
>
> ---
>
> > References:
>
> [1] Peebles _et al._ DiT: Diffusion Models with Transformer Backbones.
>
> [2] Li _et al._ TinyFusion: Layer Fusion for Diffusion Speed-up.
>
> [3] Zhao _et al._ Dynamic Diffusion Transformer.
>
> [4] Lao _et al._ DyDiT++: Dynamic Diffusion Transformers for Efficient Visual Generation.
>
> [5] Ye _et al._ Schedule On the Fly: Diffusion Time Prediction for Faster and Better Image Generation.
>
> [6] Ghosh _et al._ GenEval: An Object-Focused Framework for Evaluating Text-to-Image Alignment.
>
> [7]  Xu _et al._ ImageReward: Learning and Evaluating Human Preferences for Text-to-Image Generation.
>
> [8] Wu _et al._ Human Preference Score v2: A Solid Benchmark for Evaluating Human Preferences of Text-to-Image Synthesis.

---

> ### Comment · Reviewer_C2Gm · 2025-08-05
>
> Thanks to the detailed rebuttal. I shall maintain my positive rating.

---

> > ### Author Response · Authors · 2025-08-05
> >
> > Thank you for your time and feedback! We are glad the rebuttal was helpful and we appreciate your support for our work.

---

### Official Review · Reviewer_epHt · 2025-07-02

**Clarity:** 4
**Significance:** 3
**Originality:** 2
**Rating:** 5
**Confidence:** 3

**Summary:**

The paper proposes a novel idea for pruning T2I diffusion models, where instead of using static printing (same pruned model for all prompts as done in BKSDM) or prompt-based uniform pruning (routing prompts to specific pruned model as done in APTP), the authors propose to route each timestamp to a different pruned model based on the prompt. They use a Hypernetwork, which dynamically generates layer pruning decisions and manages timestep routing to specialized, pruned expert sub-networks. This has enabled them to make each expert cater to different groups of timestamps. Though somewhat limited, the quantitative results demonstrate the effectiveness of this method.

**Questions:**

Apart from the points listed above in "weaknesses", below are some additional questions:
- Pruning primarily happens at the final layers. Why is this the case?
- Is batch-parallelism possible with this approach? My understanding is that since each timestep can be routed to a different model, it would not be. Do the speed-up numbers take account of this?
- How much of the performance is accounted for by the final Fine-tuning after T-end? Could you include a table showing performance before and after this step?

**Ethical Concerns:**

["NO or VERY MINOR ethics concerns only"]

**Final Justification:**

The rebuttal has clarified most of my concerns and I will increase my rating to reflect this.

**Limitations:**

Yes, but only technical limitations.

**Paper Formatting Concerns:**

None.

**Quality:**

3

**Strengths And Weaknesses:**

Strengths:
+ SOTA results on the SD2.1 architecture. The performance is even better than unpruned SD2.1 (at 25/20 steps). APTP [18], the most similar related work, was unable to achieve this. At standard 50 steps, same performance can be achieved at lower MACs.
+ Clever implementation of an idea from the Language community to Vision. ToMoE [55] uses MoEs and routes each token to a different pruned subnetwork. This work does a similar thing, where each timestamp is routed to a different pruned subnetwork.
+ Comprehensive ablations have been done, and they show how different contributions impact the final results.
+ The exposition is clear and convincing.

Weaknesses:
- The overall idea is fairly close to APTP in terms of novelty. Almost similar setup is utilized, though instead of prompt-based static routing to experts, they have done prompt-based temporal routing. More insights and discussions around the similarities and differences would be helpful.
- Quantitative results are limited to a single diffusion architecture. Therefore, generalizability to other diffusion architectures is unclear.
- The choice of 10 experts as the optimal number of experts is debatable and should not be claimed as such. The experiments are only presented for SD2.1, on only 2 datasets (CLIP score is anyway not the optimal at 10 experts for MS-COCO), and just 2 metrics (FID and CLIP score). Either toning down this claim or more rigorous analysis would help.

---

> ### Author Rebuttal · Authors · 2025-07-31
>
> We thank you for the insightful review and for recognizing the strengths of our work. Below we  clarify our novelty compared to prior art and provide new results to address your questions.
>
> ---
>
> ## W1 - Novelty and Comparison to APTP
>
> We thank the reviewer for this point, and we have revised the paper to better articulate our novelty. While both methods use experts, ALTER introduces two fundamental differences from APTP: a *dynamic, temporal routing mechanism* and a **unified, single-stage training framework**.
>
> 1. **Routing Mechanism & Model Utilization:** APTP uses *static, prompt-based routing*. Each prompt is assigned to _one_ expert sub-network for all timesteps. This means an image is generated using only a static portion of the model's total capacity. In contrast, ALTER uses **dynamic, temporal routing**. Every prompt is processed by a _sequence_ of different experts, one for each timestep. This allows every image generation to leverage the **full capacity** of the model, assigning the most suitable specialized expert (e.g., "structure expert" vs. "texture expert") to each stage of the diffusion process as illustrated in Figure 1 of our paper.
> 2. **Training & Deployment Framework:** APTP requires a complex, multi-stage pipeline involving dataset clustering, experts determining and separate fine-tuning for each expert, resulting in multiple model checkpoints. ALTER is an **end-to-end framework** that learns the routing and experts jointly in a single stage, producing a single, unified model that is far simpler to manage and deploy.
>
> | Method       | Routing      | Model Usage                | Training              | Deployment               | Scalability                                       |
> | ------------ | ------------ | -------------------------- | --------------------- | ------------------------ | ------------------------------------------------- |
> | APTP         | Per-prompt   | Partial capacity per image | Multi-stage, Separate | Multiple separate models | Need re-partition & re-train for every new domain |
> | ALTER (Ours) | Per-timestep | Full capacity per image    | End-to-end, Unified   | A single unified model   | Works across domains                              |
>
> These distinctions are critical to our method's ability to improve upon the baseline's generation quality, a result not achieved by prior pruning work.
>
> ---
>
> ## W2 - Generalizability to Other Diffusion Architectures
>
> We agree that demonstrating generality is crucial. To address this, we have **conducted a new experiment applying ALTER to a DiT-XL/2 [1] backbone**. The results show that our method achieves a ~1.92x speedup while maintaining an FID score (2.36) comparable to the full model (2.27).
>
> With approximately half the layers active on average, **our method significantly outperforms static pruning (TinyFusion [2]), especially on key quality metrics like IS (254.29 vs. 234.50) and FID (2.36 vs. 2.86).** When comparing to dynamic methods like DyDiT [3], ALTER brings higher speedup. It's important to note that our approaches are **orthogonal and complementary**. ALTER performs coarse-grained **layer-level pruning**, while DyDiT performs fine-grained pruning _within_ each block.
>
> This empirically confirms that our framework is effective beyond U-Nets and is applicable to modern Transformer-based architectures.
>
> | Model                        | Speedup      | IS ($\uparrow$) | FID ($\downarrow$) | sFID $(\downarrow)$ | Precision ($\uparrow$) | Recall ($\uparrow$) |
> | ---------------------------- | ------------ | --------------- | ------------------ | ------------------- | ---------------------- | ------------------- |
> | DiT-XL/2 [1] (28-layer)      | 1.00$\times$ | 277.00          | 2.27               | 4.60                | 0.83                   | 0.57                |
> | DyDiT-XL [2] ($\lambda=0.5$) | 1.72$\times$ | 248.03          | 2.07               | 4.56                | 0.80                   | 0.61                |
> | TinyFusion [3] (14-layer)    | 1.96$\times$ | 234.50          | 2.86               | 4.75                | 0.82                   | 0.55                |
> | ALTER (Ours) (avg. 14-layer) | 1.92$\times$ | 254.29          | 2.36               | 4.63                | 0.82                   | 0.58                |
>
> ---
>
> ## W3 - The Choice of Experts Number
>
> Thank you for this advice. We agree with the reviewer that claiming 10 experts as universally "optimal" is an overstatement. Our intention was to identify a practical and effective configuration for the SDv2.1 model.
>
> We have **revised the manuscript** to tone down this claim, and refer to 10 experts as a "well-performing configuration" that demonstrates a strong efficiency-quality trade-off. Furthermore, our new experiments on the DiT architecture also successfully utilized 10 experts, achieving the strong results reported (responses to W2). This suggests that 10 is a reasonable choice across different architectures, though the ideal number may vary with the specific model and dataset.
>
> ---
>
> ## Q1 - Why does pruning primarily happen at the final layers?
>
> This is an excellent question. Our hypothesis is that it reflects the different roles of layers in the U-Net. Early/middle layers capture foundational spatial structures and low-level features that are essential throughout the entire diffusion process. Pruning them would be too destructive. In contrast, the final decoder layers are responsible for high-level semantic refinement and adding fine-grained details, tasks that are more stage-specific. Our time-aware router learns that these specialized refinement tasks can be effectively handled by smaller experts or even pruned at certain timesteps without compromising the core structure. **This observation is also consistent with prior work like APTP**, which noted that deeper layers were more amenable to pruning than the foundational ones.
>
> ---
>
> ## Q2 - Is batch-parallelism possible with this approach?
>
> Thank you for this critical question. **Yes, batch-parallelism is possible for our framework.** Our routing mechanism is deterministic, which means that for a given prompt batch and a specific timestep $t$, the routing decision is **identical for all samples in a batch**. The entire batch can therefore be processed efficiently through the same selected expert sub-network. This provides a significant practical advantage over prompt-routing methods like APTP. In APTP, different prompts within a single batch may be routed to different experts, breaking efficient batching and requiring serialization or complex batch management that can harm real-world throughput.
>
> However, to ensure fair and reproducible comparisons, our latency benchmarks in the paper follow prior work and use a batch size of 1.
>
> ---
>
> ## Q3 - How much of the performance is accounted for by the final Fine-tuning after T-end?
>
> Thanks for your suggestion. To show the contribution of the final fine-tuning stage, we have conducted the requested ablation. The results below are for our ALTER (0.65) model on MS-COCO 2017 dataset.
>
> | Model                      | Steps | CLIP ($\uparrow$) | FID-5K ($\downarrow$) |
> | -------------------------- | ----- | ----------------- | --------------------- |
> | SDv2.1                     | 20    | 31.53             | 27.83                 |
> | ALTER (0.65) (T$_{end}$)   | 20    | 31.54             | 25.98                 |
> | ALTER (0.65) (T$_{total}$) | 20    | 31.62             | 25.25                 |
> | ALTER (0.65) (T$_{end}$)   | 15    | 31.05             | 27.16                 |
> | ALTER (0.65) (T$_{total}$) | 15    | 31.19             | 26.58                 |
>
> As the results demonstrate, the model's performance after T$_{end}$ is already very strong, achieving the majority of the final quality. The final fine-tuning provides a consistent and important boost across both metrics and step counts, further improving both FID and CLIP scores.
>
> ---
>
> > References:
>
> [1] Peebles _et al._ DiT: Diffusion Models with Transformer Backbones.
>
> [2] Li _et al._ TinyFusion: Layer Fusion for Diffusion Speed-up.
>
> [3] Zhao _et al._ Dynamic Diffusion Transformer.

---

> > ### Comment · Reviewer_epHt · 2025-08-05
> >
> > The rebuttal has clarified most of my concerns and I will increase my rating to reflect this.

---

> > > ### Author Response · Authors · 2025-08-05
> > >
> > > Thank you for your response. We are glad our rebuttal could clarify your concerns. We appreciate your time and the feedback, which has helped us improve the paper.

---

### Official Review · Reviewer_F2cH · 2025-07-02

**Clarity:** 4
**Significance:** 2
**Originality:** 3
**Rating:** 5
**Confidence:** 4

**Summary:**

The paper proposes learning a hypernetwork to predict subnetworks of a pre-trained diffusion U-Net, with subnetworks mapped to certain denoising time steps. The hypernetwork is optimized in alternating steps while the U-Net is fine-tuned, and the result is a faster diffusion inference process that uses pruning but still takes advantage of the full model, pruning layers specific to each time step. Experiments are reported compared to both prior pruning strategies and caching, on SD2.1, for CC3M and COCO.

**Questions:**

Does the performance hold for other diffusion backbones? What about DiT-based architectures, which have largely replaced UNet models in modern generative pipelines?

Do more modern metrics (Image Reward) validate the results? What about human preference? It is challenging to evaluate the work with only 14 qualitative examples provided, and no human study.

**Ethical Concerns:**

["NO or VERY MINOR ethics concerns only"]

**Final Justification:**

The rebuttal addresses my concerns. I intend to keep my original rating, since I still view the submission as a solid paper and good contribution to the conference, and specifically this sub-area of diffusion acceleration. I think the motivation is sound, the method is interesting/novel, the results are good, and the methods chosen for comparison are relevant/broad.

**Limitations:**

yes

**Quality:**

3

**Strengths And Weaknesses:**

Strengths:

[S1] The method yields clear improvements over the prior pruning state-of-the-art, and beats caching works (which are somewhat comparable) as well.

[S2] The motivation follows naturally from deficiencies in prior works and is specific to the diffusion process itself.

[S3] The methodology with hypernetworks to predict the network structure is novel in this application.

Weaknesses:

[W1] The empirical evaluation is somewhat limited. More modern measurements of quality (Image Reward, Vision Reward, CLIP-IQA) are not considered, only 2 datasets are used, and, most concerningly, only 1 model is considered.

[W2] The behavior of the experts is slightly unsatisfying. Prior works in caching suggest that these earlier and middle layers can be cached, so the fact that they seem to not be able to be skipped suggests that perhaps, considering this work already computes results for every layer at some time step, perhaps implementing caching in the framework could allow for greater diversity among experts. It also seems important to investigate this behavior for additional diffusion models.

---

> ### Author Rebuttal · Authors · 2025-07-31
>
> We sincerely thank you for your detailed review and positive assessment of our work's novelty and motivation. We are pleased to present new results that address your concerns.
>
> ---
>
> ## W1 & Q2 - More Comprehensive Empirical Evaluation
>
> Thank you for suggesting a more comprehensive evaluation with modern metrics. We agree this is crucial for assessing generative quality. In response, we have benchmarked our method using **ImageReward** [1] and Human Preference Score v2 (**HPSv2**) [2], two state-of-the-art models designed to measure human aesthetic and semantic preferences. We calculate these 2 metrics on their own benchmark datasets. Since APTP is prompt-based pruning, we use the prompt split method they provided to divide the benchmark's prompts to different experts finetuned on MS-COCO and CC3M.
>
> As shown in the following two tables, the results are highly encouraging. On ImageReward, **our ALTER models rank #1 (0.65 MACs) and #2 (0.60 MACs)**, outperforming prior methods and even the original SDv2.1 baseline, which indicates a higher alignment with human preference. Similarly, on the fine-grained HPSv2 benchmark, our method consistently achieves the highest average score, demonstrating robust performance across diverse visual styles from animation to photo-realism.
>
> | Model               | MACs (G) | Image Reward ($\uparrow$) | Rank |
> | ------------------- | -------- | ------------------------- | ---- |
> | SDv2.1              | 1384.2   | 0.1139                    | 3    |
> | BK-SDM-v2 (Base)    | 876.5    | -0.0365                   | 4    |
> | APTP (0.66) (COCO)  | 916.3    | -0.8706                   | 5    |
> | APTP (0.66) (CC3M)  | 916.3    | -1.2226                   | 6    |
> | ALTER (0.65) (Ours) | 899.7    | 0.1494                    | 1    |
> | ALTER (0.60) (Ours) | 830.5    | 0.1487                    | 2    |
>
> | Model               | MACs (G) | HPSv2 ($\uparrow$) | Rank |
> | ------------------- | -------- | ------------------ | ---- |
> | SDv2.1              | 1384.2   | 24.47              | 2    |
> | BK-SDM-v2 (Base)    | 876.5    | 23.97              | 4    |
> | APTP (0.66) (COCO)  | 916.3    | 19.08              | 5    |
> | APTP (0.66) (CC3M)  | 916.3    | 18.19              | 6    |
> | ALTER (0.65) (Ours) | 899.7    | 24.54              | 1    |
> | ALTER (0.60) (Ours) | 830.5    | 24.36              | 3    |
>
> ---
>
> ## W2 - Behavior of the Experts
>
> This is an excellent question that makes us clarify the relationship between our dynamic pruning method and caching. The core difference is that caching methods reuse a layer's **output features** from a previous step, whereas our method, ALTER, learns to skip the **entire computation** of a layer.
>
> Caching methods like DeepCache [3] show that the _outputs_ of early-to-mid layers can be reused across steps, as they capture low-frequency information that evolves slowly.  However, our results show that foundational early and middle layers are rarely pruned, suggest that while their outputs might be stable enough to be cached, the computations performed within them are still **essential at every step** to progressively refine the feature maps for later layers. Skipping these layers entirely is too detrimental to the final image quality. This observation is also consistent with prior work like APTP, which noted that deeper layers were more amenable to pruning than the foundational ones.
>
> We agree that encouraging greater routing diversity is a valuable direction for future work, potentially using regularization techniques. We also strongly agree that caching methods could **complement** ALTER. This highlights a powerful scheme: ALTER could first prune the model's architecture, and a caching scheme could then be applied to the remaining active layers for further speedups.
>
> ---
>
> ## Q1 - Does the performance hold for other diffusion backbones?
>
> To empirically validate that our framework generalizes beyond U-Nets, we have **conducted a new experiment applying ALTER to a DiT-XL/2 [4] backbone** on ImageNet (256 $\times$ 256 resolution generation, 250 steps). For the DiT architecture, each Transformer block was treated as a prunable unit, and the ALTER framework was applied without fundamental architectural changes.
>
> | Model                        | Speedup      | IS ($\uparrow$) | FID ($\downarrow$) | sFID $(\downarrow)$ | Precision ($\uparrow$) | Recall ($\uparrow$) |
> | ---------------------------- | ------------ | --------------- | ------------------ | ------------------- | ---------------------- | ------------------- |
> | DiT-XL/2 [4] (28-layer)      | 1.00$\times$ | 277.00          | 2.27               | 4.60                | 0.83                   | 0.57                |
> | DyDiT-XL [5] ($\lambda=0.5$) | 1.72$\times$ | 248.03          | 2.07               | 4.56                | 0.80                   | 0.61                |
> | TinyFusion [6] (14-layer)    | 1.96$\times$ | 234.50          | 2.86               | 4.75                | 0.82                   | 0.55                |
> | ALTER (Ours) (avg. 14-layer) | 1.92$\times$ | 254.29          | 2.36               | 4.63                | 0.82                   | 0.58                |
>
> With approximately half the layers active on average, **our method significantly outperforms static pruning (TinyFusion [6]), especially on key quality metrics like IS (254.29 vs. 234.50) and FID (2.36 vs. 2.86). It is nearly comparable with the full DiT-XL model,** demonstrating a good quality-efficiency trade-off. When comparing to dynamic methods like DyDiT [5], ALTER brings higher speedup. It's important to note that our approaches are **orthogonal and complementary**. ALTER performs coarse-grained **layer-level pruning**, while DyDiT performs fine-grained pruning _within_ each block.
>
> This successful application of ALTER to a DiT backbone and its orthogonality to fine-grained methods, validates that ALTER is a general and effective framework for accelerating various diffusion architectures.
>
> ---
>
> > References:
>
> [1] Xu _et al._ ImageReward: Learning and Evaluating Human Preferences for Text-to-Image Generation.
>
> [2] Wu _et al._ Human Preference Score v2: A Solid Benchmark for Evaluating Human Preferences of Text-to-Image Synthesis.
>
> [3] Ma _et al._ DeepCache: Accelerating Diffusion Models for Free.
>
> [4] Peebles _et al._ DiT: Diffusion Models with Transformer Backbones.
>
> [5] Zhao et al. Dynamic Diffusion Transformer.
>
> [6] Li _et al._ TinyFusion: Layer Fusion for Diffusion Speed-up.

---

> > ### Comment · Reviewer_F2cH · 2025-07-31
> > **Concerns Resolved**
> >
> > This resolves my concerns. I still do not think the paper quite rises to the level of strong accept, but it is a very solid submission and above the bar for acceptance.

---

> > > ### Author Response · Authors · 2025-08-01
> > >
> > > Thank you for your feedback! We truly appreciate your time and effort in reviewing our paper. Thank you again for your valuable comments to improve the quality of our work!

---

### Official Review · Reviewer_jkuL · 2025-07-03

**Clarity:** 4
**Significance:** 3
**Originality:** 3
**Rating:** 5
**Confidence:** 4

**Summary:**

The paper proposes a novel method, **ALTER**, which enhances the efficiency of diffusion models by introducing an innovative combination of layer pruning and temporal expert routing. The method aims to reduce computational overhead during inference without sacrificing generative quality. Key contributions include the unification of layer pruning, expert routing, and model fine-tuning into a single-stage optimization framework. This results in a significant reduction in the MACs and inference time while maintaining the visual fidelity of generated images. The experiments show that ALTER outperforms existing methods, such as static pruning and sample-wise dynamic pruning, on standard datasets.

**Questions:**

- In Tables 1 and 2, the authors use FID, CLIP, and CMMD to evaluate the generation quality. While these are commonly used metrics, considering that this paper focuses on the acceleration of diffusion tasks, the information deviation before and after acceleration should also be considered as a metric. Therefore, adding SSIM and PSNR to measure the generation deviation from the pre-trained model could further enhance the evaluation of generation quality.

- In principle, the ALTER method has good scalability. It is orthogonal to the current Diffusion Cache and token reduction methods. In the future, further validation on its scalability could be conducted, which would make it even more comprehensive.

- Additionally, I have a small suggestion. I recommend changing "Temporal Expert Routing" to "Step Expert Routing," as it might be easier to understand. The term "Step" could have a higher acceptance rate among those familiar with diffusion models, although "Temporal" is also fine.

- Some symbols in the paper are used redundantly. For example, the symbol t represents both the timestep of the diffusion model and the training step in Algorithm 1. It would be helpful to standardize the notation throughout the paper.

- Overall, the idea presented in this paper is very interesting. If the concerns mentioned above can be addressed, I would consider raising my score.

**Ethical Concerns:**

["NO or VERY MINOR ethics concerns only"]

**Final Justification:**

The authors have addressed all previously raised concerns thoroughly. The evaluation has been enhanced by incorporating SSIM and PSNR, offering a more complete assessment of the generation deviation introduced by acceleration. This addition strengthens the empirical support for the proposed method.

The scalability of the ALTER framework has been further validated, confirming its compatibility with existing techniques such as Diffusion Cache and token reduction methods. These updates underline the method’s generality and practical relevance.

Overall, the paper now presents a clear, well-structured, and comprehensive contribution to efficient diffusion model inference. It demonstrates both technical soundness and practical value. I recommend acceptance.

**Limitations:**

The authors have adequately addressed the limitations of the model.

**Paper Formatting Concerns:**

No significant formatting issues were noted, and the paper follows the NeurIPS formatting guidelines well.

**Quality:**

3

**Strengths And Weaknesses:**

**Strengths**

- The integration of layer pruning with temporal expert routing is a novel approach, offering flexibility and better utilization of model capacity across different time steps in the diffusion process.

- The method achieves a substantial speedup (e.g., 3.64x faster than the original Stable Diffusion v2.1) while maintaining competitive generative quality, making it a strong candidate for deployment in resource-constrained environments.

- Extensive experiments across various datasets (e.g., CC3M, MS-COCO) demonstrate the effectiveness of the proposed method. The paper also includes ablation studies, showing the contribution of key components like the temporal router and joint optimization.

- The paper is well-organized and easy to understand.

**Weaknesses**

Please refer to the Questions part.

---

> ### Author Rebuttal · Authors · 2025-07-31
>
> We sincerely thank you for your constructive feedback and for recognizing the novelty and solid empirical results of ALTER. Below we address each question in turn.
>
> ---
>
> ## Q1 - Add SSIM and PSNR to measure the generation deviation from the pre-trained model
>
> We provide the SSIM and PSNR to show the deviation of different methods from the original SDv2.1 25 step inference.
>
> | Model               | Steps | MACs (G) ($\downarrow$) | Latency (s) ($\downarrow$) | CC3M SSIM ($\uparrow$) | CC3M PSNR ($\uparrow$) | COCO SSIM ($\uparrow$) | COCO PSNR ($\uparrow$) |
> | ----------| ----- | ----------------------- | -------------------------- | ---------------------- | ---------------------- | ------------------------- | ------------------------- |
> | SDv2.1              | 25    | 1384.2                  | 4.0                        | -                      | -                      | -                         | -                         |
> | SDv2.1              | 20    | 1384.2                  | 3.2                        | 0.3529                 | 13.71                  | 0.1582                    | 10.24                     |
> | BK-SDM-v2 (Base)    | 25    | 876.5                   | 2.5                        | 0.1208                 | 9.42                   | 0.1551                    | 9.47                      |
> | APTP (0.66)         | 25    | 916.3                   | 2.6                        | 0.0782                 | 8.99                   | 0.0730                    | 8.83                      |
> | ALTER (0.65) (Ours) | 25    | 899.7                   | 2.6                        | 0.1233                 | 9.90                   | 0.3261                    | 12.81                     |
> | ALTER (0.65) (Ours) | 20    | 899.7                   | 2.1                        | 0.1247                 | 9.95                   | 0.2484                    | 11.61                     |
> | ALTER (0.60) (Ours) | 20    | 830.5                   | 1.9                        | 0.1224                 | 9.94                   | 0.1898                    | 10.73                     |
>
> As shown in the above table, our method **ALTER not only matches the MACs of prior work but also achieves significantly higher SSIM and PSNR scores**, which demonstrates that our approach better preserves the generative characteristics of the original model.
>
> To provide an even more comprehensive evaluation of generation quality, we also benchmarked our models using advanced preference models, **ImageReward** [1] and Human Preference Score v2 (**HPSv2**) [2] (all at 25 steps). These metrics assess perceptual quality and human aesthetic preference. As shown below, **our method ranks highest on both metrics**, even outperforming the original SDv2.1, confirming that our efficiency gains do not come at the cost of visual quality.
>
> | Model               | MACs (G) | Image Reward ($\uparrow$) | HPSv2 ($\uparrow$) |
> | ------------------- | -------- | ------------------------- | ------------------ |
> | sd2.1               | 1384.2   | 0.1139                    | 24.47              |
> | BK-SDM-v2-Base      | 876.5    | -0.0365                   | 23.97              |
> | APTP-Small (COCO)   | 916.3    | -0.8706                   | 19.08              |
> | APTP-Small (CC3M)   | 916.3    | -1.2226                   | 18.19              |
> | ALTER (0.65) (Ours) | 899.7    | 0.1494              | 24.54          |
> | ALTER (0.60) (Ours) | 830.5    | 0.1487                    | 24.36              |
>
> ---
>
> ## Q2 - Scalability / Orthogonality to other accelerators
>
> We appreciate the reviewer for highlighting this key advantage of our work. We strongly agree that ALTER’s design makes it orthogonal to and compatible with other diffusion acceleration methods such as Diffusion Cache and token reduction.
>
> Here we conduct a new experiment combining ALTER with a token reduction method ToMeSD [3] to demonstrate ALTER's scalability. For this combination, we follow the default setting of ToMeSD. As shown in the following table, we **boost the total speedup from 3.64x to an impressive 6.54x** with only a minor, controllable trade-off in performance scores. This empirically validates the excellent scalability of our method.
>
> | Model               | Steps | ToMe r% | Speedup      | CLIP ($\uparrow$) | FID-5K ($\downarrow$) |
> | ------------------- | ----- | ------- | ------------ | ----------------- | --------------------- |
> | SDv2.1              | 50    | 0       | 1.00$\times$ | 31.55             | 27.29                 |
> | SDv2.1              | 20    | 0       | 2.49$\times$ | 31.53             | 27.83                 |
> | ALTER (0.65) (Ours) | 20    | 0       | 3.64$\times$ | 31.62             | 25.25                 |
> | w/ ToMeSD           | 20    | 40      | 4.93$\times$ | 31.48             | 25.93                 |
> |                     | 20    | 50      | 5.75$\times$ | 31.34             | 26.89                 |
> |                     | 20    | 60      | 6.54$\times$ | 31.24             | 27.96                 |
>
> Regarding caching methods like DeepCache [4], we agree this is another exciting direction. Since ALTER dynamically prunes layers, a compatible cache would only need to store features and reuse for current active layers, potentially enabling a more memory-efficient caching scheme. We will leave this as future work.
>
> ---
>
> ## Q3 - Rename "Temporal Expert Routing" to "Step Expert Routing"
>
> Thank you for this suggestion. We appreciate you pointing out that "Step Expert Routing" might be more immediately familiar to some readers in the diffusion community. We have carefully considered this change. However, we have a deliberate reason for our choice of "Temporal" and would prefer to retain it. Our motivation is that the routing mechanism is conditioned on the timestep variable $t$ of the diffusion process, which is represented by the _time embedding_. While it is discretized into steps during diffusion sampling, our framing is rooted in the underlying temporal nature of the process.
>
> ---
>
> ## Q4 - Standardize symbol usage
>
> Thank you for your careful reading and pointing out this redundancy. We have clearly checked the notation definition and standardized all notation in the revised manuscript. We now use $t$ exclusively for the diffusion timestep and have introduced $k$ to denote the training iterations.
>
> ---
>
> > References:
>
> [1] Xu _et al._ ImageReward: Learning and Evaluating Human Preferences for Text-to-Image Generation.
>
> [2] Wu _et al._ Human Preference Score v2: A Solid Benchmark for Evaluating Human Preferences of Text-to-Image Synthesis.
>
> [3] Bolya _et al._ Token Merging for Fast Stable Diffusion.
>
> [4] Ma _et al._ DeepCache: Accelerating Diffusion Models for Free.

---

> > ### Comment · Reviewer_jkuL · 2025-08-01
> >
> > Thank you for your response. My concerns have been effectively resolved, and I'm willing to increase my score to 5.

---

> > > ### Author Response · Authors · 2025-08-01
> > >
> > > Thank you for your time and effort throughout the review process! We are glad that our response has effectively resolved your concerns. We also appreciate your valuable suggestions for future work.

---

### Decision · Program_Chairs · 2025-09-17

**Decision:**

Accept (poster)

**Comment:**

This paper introduces ALTER, a unified framework for improving the efficiency of diffusion models by jointly optimizing layer pruning, expert routing, and model fine-tuning in a single stage. Unlike prior uniform or sequential pruning strategies, ALTER employs a trainable hypernetwork to dynamically generate pruning decisions and route timesteps to specialized pruned experts during fine-tuning. This co-optimization achieves substantial efficiency gains—delivering a 3.64× speedup with 35% sparsity and only 25.9% of the original MACs—while maintaining the visual fidelity of the original diffusion model.

The reviewers agreed on the importance of the problem, the novelty of the approach, and the strength of the empirical results. They also highlighted the significant speedups achieved across benchmarks and the clarity of the presentation. For these reasons, we recommend acceptance.